# Integrin-mediated electric axon guidance underlying optic nerve formation in the embryonic chick retina

Masayuki Yamashita [1✉]

Retinal ganglion cell (RGC) axons converge on the optic disc to form an optic nerve. However, the mechanism of RGC axon convergence remains elusive. In the embryonic retina, an electric field (EF) exists and this EF converges on the future optic disc. EFs have been demonstrated in vitro to orient axons toward the cathode. Here, I show that the EF directs RGC axons through integrin in an extracellular $Ca^{2+}$-dependent manner. The cathodal growth of embryonic chick RGC axons, which express integrin α6β1, was enhanced by monoclonal anti-chicken integrin β1 antibodies. $Mn^{2+}$ abolished these EF effects, as $Mn^{2+}$ occupies the $Ca^{2+}$-dependent negative regulatory site in the β1 subunit to eliminate $Ca^{2+}$ inhibition. The present study proposes an integrin-mediated electric axon steering model, which involves directional $Ca^{2+}$ movements and asymmetric microtubule stabilization. Since neuroepithelial cells generate EFs during neurogenesis, electric axon guidance may primarily be used in central nervous system development.

[1] International University of Health and Welfare, Ohtawara, Japan. ✉email: my57@iuhw.ac.jp

During the development of the central nervous system (CNS) including the retina, the first-born neurons extend long-distance traveling axons, such as commissural axons and optic nerves, although the guidance cue for these long, first axons remains unknown. The prevailing belief is currently that growth cones detect concentration gradients of long-range chemoattractants. However, gradients alone are unlikely to provide a directional cue with high fidelity because of fundamental physical limits on gradient sensing[1]. In fact, netrin was initially proposed to be a diffusible long-range chemoattractant for commissural axons, but later studies revealed that it acts locally by promoting growth cone adhesion in the brain and spinal cord[2], or as a short-range cue at the optic disc[3].

Growing axons are directed not only by chemical signals but also by electric fields (EFs) in a process called galvanotropism[4]. The first experimental evidence for this phenomenon was reported over a century ago, in 1920[5], after the invention of the tissue culture technique[6]. However, the physiological relevance of electric axon guidance remains controversial for two reasons, (1) because the in vivo evidence for its importance has been scant and (2) because the molecular mechanisms are somewhat unclear.

The retina is a suitable model for studying the development of the CNS. In the embryonic chick retina, there is an extracellular voltage gradient, which is generated by sodium transport of neuroepithelial cells[7,8]. This EF converges on the future optic disc and is essential for the correct orientation of RGC axons[7]. Thus, the possibility was raised that the EF acts as a guidance cue for RGC axons. This possibility was examined in the present study by using the embryonic chick retina. The molecular mechanisms were also studied based on the following considerations.

An EF causes directional movements of ions in the extracellular space. These ionic movements can be a directional cue, because EF-moved ions encounter cell surface molecules differently on the anodal and cathodal sides of an axon. The moving ions may asymmetrically regulate the cell surface molecule that triggers axon steering. As a candidate molecule for this role, integrin was considered for the following reasons: (1) embryonic chick RGC axons express integrin $\alpha6\beta1$[9], (2) the affinity of integrin for the extracellular matrix (ECM) ligand is dynamically regulated by the extracellular $Ca^{2+}$ in the physiological concentration range[10,11], (3) the loss of integrin results in disorganization of RGC axons[12], (4) integrin mediates EF-induced directional cell migration[13–16], and (5) a reduction in extracellular $Ca^{2+}$ acts as a guidance cue for integrin-mediated cell migration during wound healing[17]. The present study demonstrates that the EF orients RGC axons through integrin and proposes a $Ca^{2+}$-dependent axon steering model. The role of EFs in CNS development is discussed.

## Results

**RGC axons extend toward the cathode**. Retinal strips of embryonic day 6 (E6) chicks were explanted from the segment that was originally dorsal to the optic nerve head (Fig. 1a, b). They were embedded in Matrigel® because Matrigel® and the optic fiber layer contain the ECM ligand for integrin[18,19]. Since the orientation of RGC axons growing out from the retinal strip mirrors the pattern of axon growth during normal development in vivo[20], numerous axons emerged from the ventral side of retinal strips, as shown by the fluorescence imaging of live axons (Fig. 1c, e). It was likely that the axons that had already been oriented before the explant extended ventrally[20,21]. The RGC axons appeared preferentially at the central part of retinal strips, where the density of RGCs is high[22]. The ventral extension was quantified from the fluorescence intensity (ventral extension rate, VER, Fig. 1c, d).

Growing axons are directed by EF toward the cathode[4,21,23,24]. The ventral extension increased in the ventrally directed EF, of which strength was kept at 15 mV/mm to mimic the in vivo EF[7] (Fig. 1f, k). Conversely, it decreased in the dorsally directed EF (Fig. 1j, k), which is known as anodal suppression[23]. These results may imply that the EF directed the axons of newborn RGCs or unoriented axons toward the cathode and could also reorient the ventrally growing axons. The dorsal axons appeared to increase in the reversal EF, although they were not analyzed because the background intensity had to be raised to visualize them (Fig. 1m).

**Integrin mediates EF effects**. To examine whether integrin was involved in the electric effect, retinal strips were preincubated in culture medium containing the monoclonal anti-chicken integrin $\beta1$ antibody TASC[25] and then embedded in Matrigel® containing TASC. The ventral extension toward the cathode was enhanced in a dose-dependent manner (Fig. 1g, k, l). Another monoclonal anti-chicken integrin $\beta1$ antibody, W1B10[26] also enhanced cathodal growth more effectively than TASC at low concentrations (≤50 μg/mL, Fig. 1h, k, l). The negative control isotype antibody mouse IgG1 did not enhance cathodal growth even at a high concentration (200 μg/mL, Fig. 1i, k). TASC and W1B10 also increased the ventral extension without EF (Supplementary Fig. 1b–d), while the negative control isotype antibody mouse IgG1 did not increase the ventral extension without EF (Supplementary Fig. 1a, d). Despite the fact that TASC and W1B10 increased the basal extension, the increases in VER with EF compared to without EF [(VERs with EF) − (mean VER without EF), EF index, EFI] were larger than the control without the antibodies (Supplementary Fig. 1e). Thus, integrin mediated EF effects, but a possibility remained that integrin might have regulated axon extension itself without being involved in axon orientation. To examine whether integrin was involved in the electric orientation of axons, the role of $Ca^{2+}$ in integrin regulation was investigated in the following sections.

**$Mn^{2+}$ abolishes EF effects**. The ligand-binding domain of the integrin $\beta1$ subunit contains the $Ca^{2+}$-dependent negative regulatory site termed ADMIDAS[11]. $Ca^{2+}$ binding to ADMIDAS changes integrin into the closed conformation to inhibit ligand binding[11]. As the EF moves $Ca^{2+}$ in the gelatinous ECM, it is possible that the directionally moved $Ca^{2+}$ regulates integrin asymmetrically at the axon surface. To examine this possibility, $Mn^{2+}$ was added to the preincubation medium, Matrigel®, and the culture medium because $Mn^{2+}$ occupies ADMIDAS and stabilizes integrin in the open conformation to maintain high integrin-ligand affinity[11]. $Mn^{2+}$ at 500 μM abolished the electric effect even in the presence of TASC or W1B10 (Fig. 2a–g, Supplementary Fig. 2a).

When the concentration of $Mn^{2+}$ was raised to 1.0 mM (eleven retinae tested) and 1.5 mM (five retinae tested), the ventral outgrowth was suppressed (Supplementary Fig. 2b). Instead, local arborizations were found within the retinal strip (Supplementary Fig. 2c). These results of high $Mn^{2+}$ may imply that axon extension requires the dissociation of integrin from the ligand (discussed later).

**Correct orientation of RGC axons by EF**. To examine whether the $Ca^{2+}$ regulation of integrin underlies the correct orientation of RGC axons in vivo, $Mn^{2+}$ was applied to the organotypic culture of the whole neural retina from E4 chicks. In the control retinae incubated for 24 h without $Mn^{2+}$, RGC axons were correctly oriented at the central region near the optic nerve head and outside the central region[27] (Fig. 2h, i, four retinae tested). In the culture medium containing 500 μM $Mn^{2+}$, the axons close to the

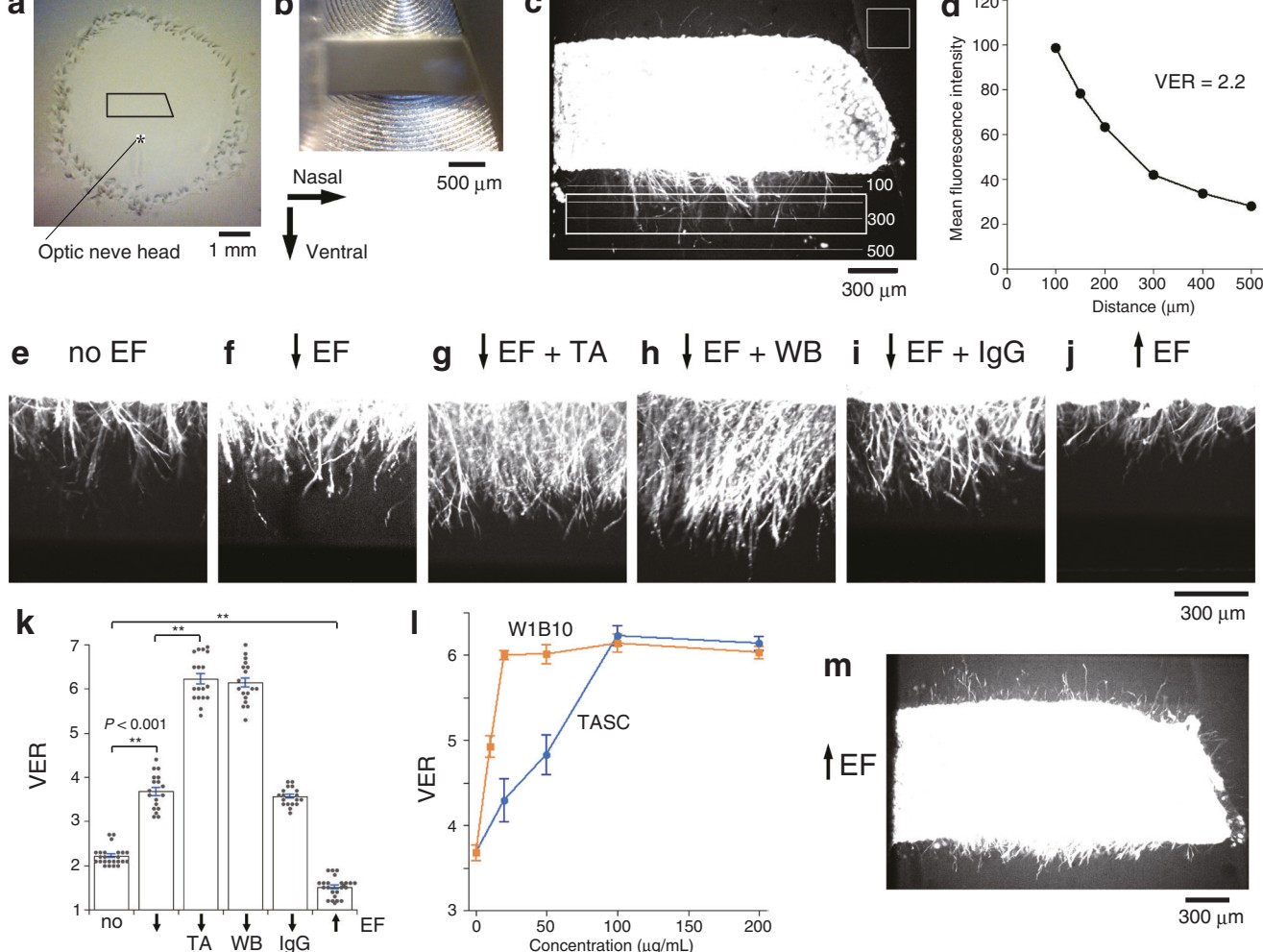

**Fig. 1 EFs direct RGC axons toward the cathode and integrin β1 antibodies enhance the EF effect. a** A retinal strip was explanted from the segment dorsal to the optic nerve head of embryonic day 6 (E6) chick retina. **b** The retinal strip was fixed with a wedge-shaped membrane filter inserted between the oblique nasal edge of the retinal strip and the wall of culture chamber. The temporal edge makes contact with the opposite wall. **c** A retinal strip incubated for 24 h without EF. Live cells and RGC axons were fluorescently labeled with calcein-AM. RGC axons extended from the ventral edge of the retinal strip. Horizontal lines indicate the distance from the ventral edge. **d** The mean fluorescence intensity measured at the lines in (**c**) was plotted against the distance from the ventral edge. Ventral extension rate (VER) is the ratio of the mean fluorescence intensity between the lines 150 μm and 400 μm ventral to the retinal strip indicated by the rectangle in (**c**) against the background intensity measured at the area without the retina indicated by the square at the upper right in (**c**). **e** RGC axons extending from the ventral edge at the central part of retinal strip in (**c**). **f** RGC axons cultured in the ventrally directed EF (downward arrow). **g** RGC axons cultured with EF and TASC (100 μg/mL). **h** RGC axons cultured with EF and W1B10 (100 μg/mL). **i** RGC axons cultured with EF and the control antibody IgG1 (200 μg/mL). **j** RGC axons cultured in the dorsally directed EF (upward arrow). **k** VERs (mean ± s.e.m.) of retinal strips cultured (from left), without EF (no, n = 24 photos taken at different focus levels from 4 retinal strips); with the ventrally directed EF (downward arrow, n = 18 from 3 retinal strips); with EF and TASC (100 μg/mL, n = 18 from 3 retinal strips); with EF and W1B10 (100 μg/mL, n = 18 from 3 retinal strips); with EF and the control antibody (200 μg/mL, n = 18 from 3 retinal strips); with the dorsally directed EF (upward arrow, n = 24 from 4 retinal strips). Horizontal bars and asterisks denote significant differences (two-tailed t-test, **P < 0.001). **l** Dose–response relationships between the concentration of antibodies (TASC, W1B10) and VER (mean ± s.e.m.) of retinal strips cultured in the ventrally directed EF (n = 18 from 3 retinal strips for each point). VERs at 0 and 100 μg/mL were replotted from (**k**). **m** A retinal strip cultured in the dorsally directed EF. Background intensity was raised to visualize the axons on the dorsal side. The axons on the ventral side are shown in (**j**) at normal background level.

optic nerve head were oriented correctly (Fig. 2j), whereas chaotic extensions were found outside the central region (Fig. 2k, yellow arrowheads, seven retinae tested). These abnormal trajectories might have been due to a deleterious effect of $Mn^{2+}$. However, the fact that RGC axons had normally extended at the central region could exclude such an effect. Since RGC axons first develop at the central region[27], the axons near the optic nerve head seemed to have already been correctly oriented before the addition of $Mn^{2+}$. Thus, $Ca^{2+}$ binding to ADMIDAS was considered to be essential for the correct orientation of RGC axons

in vivo. The same results were observed in the previous study[7], where the endogenous EF was suppressed by applying amiloride to the organotypic culture to block epithelial sodium channels of neuroepithelial cells[8]. From the previous and present results, it seemed likely that the endogenous EF directs RGC axons through integrin in an extracellular $Ca^{2+}$-dependent manner.

Assuming that the endogenous EF directs RGC axons to the optic nerve head, the stem of an optic nerve could be reproduced in vitro by focusing the EF. To test this idea, a retinal strip was placed in front of a fan-shaped microfluidic chip (Supplementary

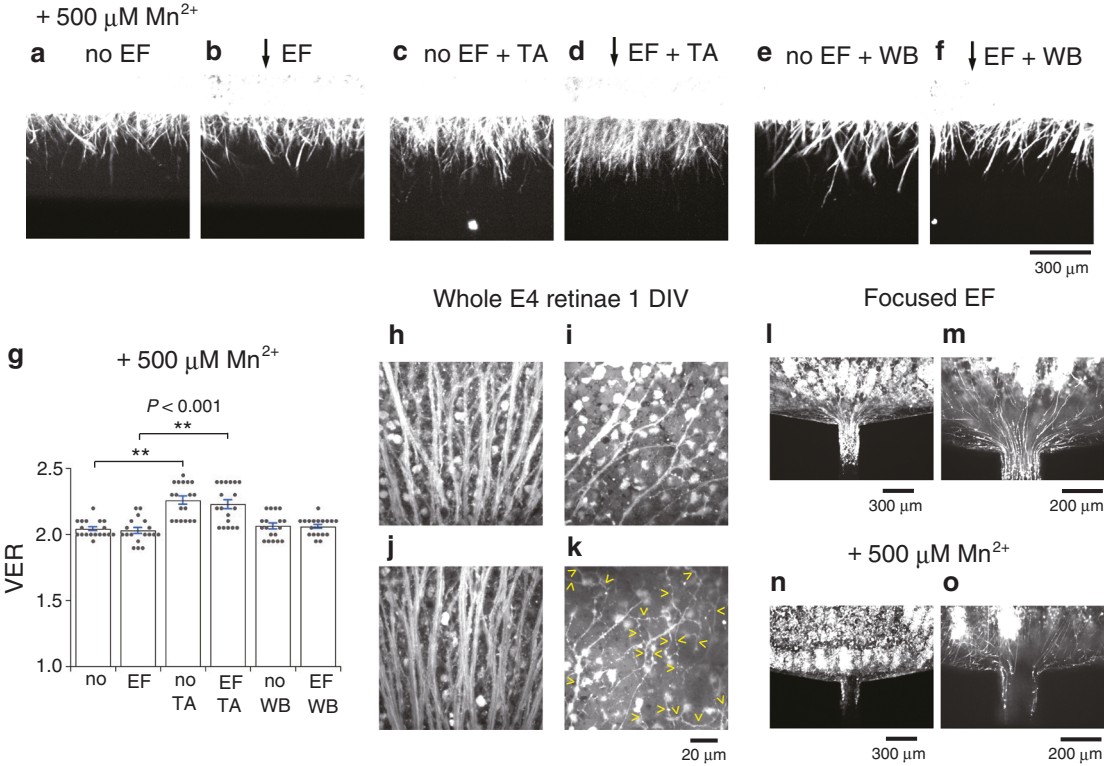

**Fig. 2 Mn$^{2+}$ abolishes EF effects and EF is essential for correct orientation of RGC axons. a** RGC axons cultured with 500 µM Mn$^{2+}$. **b** RGC axons cultured with 500 µM Mn$^{2+}$ in the ventrally directed EF. **c** RGC axons cultured with 500 µM Mn$^{2+}$ and TASC (100 µg/mL). **d** RGC axons cultured with 500 µM Mn$^{2+}$ and TASC (100 µg/mL) in the ventrally directed EF. **e** RGC axons cultured with 500 µM Mn$^{2+}$ and W1B10 (100 µg/mL). **f** RGC axons cultured with 500 µM Mn$^{2+}$ and W1B10 (100 µg/mL) in the ventrally directed EF. **g** VERs (mean ± s.e.m.) of retinal strips cultured with 500 µM Mn$^{2+}$ (from left), without EF (no); with the ventrally directed EF (EF); without EF and with TASC (no/TA); with EF and TASC (EF/TA); without EF and with W1B10 (no/WB); with EF and W1B10 (EF/WB). Each column represents the value obtained from 18 photos taken at different focus levels from 3 retinal strips. Horizontal bars and asterisks denote significant differences (two-tailed *t*-test, **$P < 0.001$). **h**, **i** A whole E4 retina cultured without Mn$^{2+}$ for 24 h (one day in vitro, 1 DIV). **h** RGC axons in the central region dorsal to the optic nerve head. **i** RGC axons in the dorso-nasal region extending to the optic nerve head (to the lower left). **j**, **k** A whole E4 retina cultured with 500 µM Mn$^{2+}$ for 24 h. **j** RGC axons in the central region dorsal to the optic nerve head. **k** RGC axons in the dorso-nasal region extending randomly. Yellow arrowheads point to abnormal axon trajectories. **l**, **m** A retinal strip cultured in a focused EF. **l** The ventral side of the retinal strip facing the open side of a fan-shaped microfluidic chip with the central channel, through which a current of 56 µA flowed to the cathode. The outgrowing axons converged on the channel and entered it. **m** A high-magnification image of the converging axons in (**l**). **n**, **o** A retinal strip cultured with 500 µM Mn$^{2+}$ in the same focused EF. **n** The ventral side of the retinal strip. **o** A high-magnification image of (**n**). The EF strength by applying 56 µA was estimated as 9 mV/mm around the retinal strip and that at the entrance of the channel was 300 mV/mm (see "Methods").

Fig. 3), of which the central channel (0.2 mm in width) allowed the current to flow to the cathode. In the focused EF, RGC axons converged on and entered the channel (five retinae tested at 56 µA, Fig. 2l, m; two retinae tested at 6.0 µA and 0.6 µA, respectively, Supplementary Fig. 4a–d). Even without the exogenous current, RGC axons gathered and entered the channel by connecting the channel to the culture medium of excess volume, such as electrical grounding (two retinae tested, Supplementary Fig. 4e, f). This fact could be explained by the active ion transport of the retinal neuroepithelium itself (see Discussion). When the exit of the channel was blocked with a rubber plug to prevent any ionic flows, RGC axons did not converge (three retinae tested, Supplementary Fig. 4g, h). Mn$^{2+}$ also abolished axon convergence (three retinae tested at 56 µA, Fig. 2n, o). These results strongly support the idea that RGC axons are electrically directed to the future optic disc[7].

**Integrin mediates RGC axon convergence**. To quantify RGC axon convergence in focused EFs, a microchannel chamber was designed (Supplementary Fig. 5). In the focused EF by applying a current of 120 µA, RGC axons converged on the microchannel (Fig. 3a). The convergence angle (CA) was measured on the axons that had extended straight from the nasal and temporal regions toward the microchannel (Supplementary Fig. 6a, Fig. 3d). The fluorescence intensities within CA were measured at the half-distance line between the retina and the microchannel (Supplementary Fig. 6a, d). The total sum of these fluorescence intensities was calculated as another parameter of axon convergence (F$_{CA}$, Fig. 3e). Furthermore, F$_{CA}$ was divided by the number of pixels of the half-distance line to obtain mean fluorescence intensity at one pixel as an index of axon density (mean F$_{CA}$, Fig. 3f). To determine whether integrin mediated RGC axon convergence, W1B10 and Mn$^{2+}$ were tested. W1B10 (100 µg/mL) increased CA, F$_{CA}$, and mean F$_{CA}$ (Fig. 3b, d–f, Supplementary Fig. 6b, e), whereas Mn$^{2+}$ at 100 µM decreased them (Fig. 3c, d–f, Supplementary Fig. 6c, f). Without applying currents, RGC axons extended without convergence (three retinae tested, Supplementary Fig. 6g). Mn$^{2+}$ at 500 µM completely abolished axon convergence (three retinae tested, Supplementary Fig. 6h). These results suggested that integrin mediated electric orientation of RGC axons and their convergence in the focused EF.

**Asymmetry of EF-moved Ca$^{2+}$**. To reveal Ca$^{2+}$ dynamics around an axon in an EF, Ca$^{2+}$ movements were imaged at the

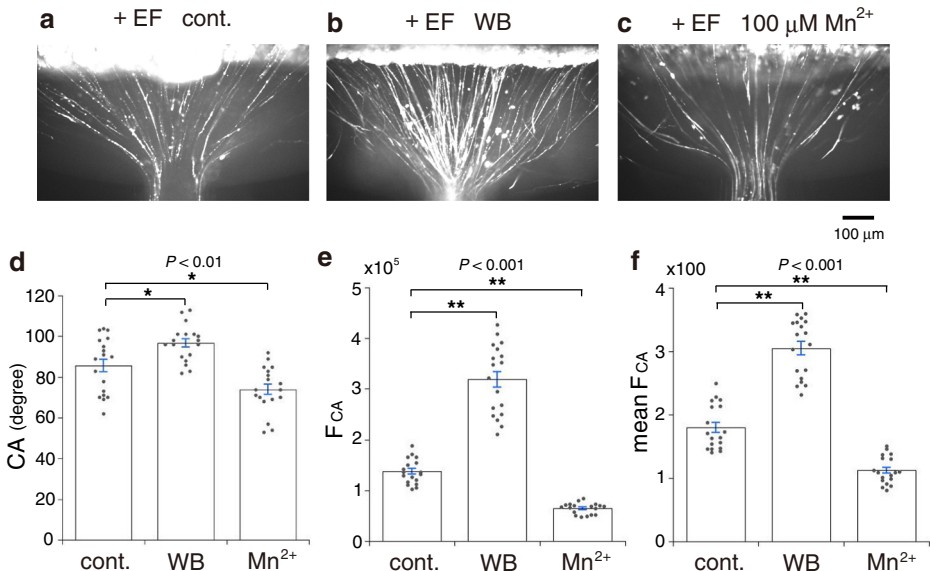

**Fig. 3 RGC axon convergence in microchannel chamber. a–c** RGC axons extending from the ventral edge of retinal strips cultured in a focused EF by applying a current of 120 μA. **a** RGC axons cultured without drug. **b** RGC axons cultured with W1B10 (100 μg/mL). **c** RGC axons cultured with 100 μM Mn²⁺. **d** Convergence angle (CA, mean ± s.e.m.) of RGC axons cultured in the focused EF without drug (cont.), with W1B10 (WB), and 100 μM Mn²⁺ (Mn²⁺). CA is the angle between the axons extending straight from the nasal and temporal regions of retinal strip (yellow lines in Supplementary Fig. 6a–c). Each column represents the value obtained from 18 photos taken at different focus levels from 3 retinal strips. **e** Total of the fluorescence intensities within CA ($F_{CA}$, mean ± s.e.m.). The fluorescence intensities were measured at the half-distance line between the retina and the microchannel (cyan lines in Supplementary Fig. 6a–c). Their total sum was calculated from the transverse profiles (Supplementary Fig. 6d–f). Data were obtained from the photos used in (**d**). **f** Mean fluorescence intensities within CA (mean $F_{CA}$, mean ± s.e.m.). $F_{CA}$ was divided with the number of pixels of the half-distance line. Horizontal bars and asterisks denote significant differences (two-tailed *t*-test, \*$P < 0.01$, \*\*$P < 0.001$). The EF strength by applying 120 μA was estimated as 20 mV/mm around the retinal strip and that at the entrance of the microchannel was 300 mV/mm (see "Methods").

anodal and cathodal sides of an axon. For this purpose, an anionic Ca²⁺-sensitive fluorescent dye (Calbryte^TM−520L, Kd: 90 μM) was used in the Matrigel®-based thin layer dissociated culture. A single axon was identified on the high-magnification confocal microscope system with transmitted light (Supplementary Fig. 7). Fluorescence was measured on the axons that were perpendicular to the direction of EF since it was supposed that the axon surface would act as a screen for EF-moved Ca²⁺. Growth cones were not selected for this measurement because their fine protrusions were not identified with transmitted light. To avoid the saturation of Ca²⁺-sensitive fluorescence, the Matrigel® thin layer was washed with a Ca²⁺-free solution. A micropipette containing 1 mM CaCl₂ and 10 mM Calbryte^TM−520L was positioned at the anodal side of the axon (Fig. 4a). Ca²⁺ diffused out of the pipette into the low-Ca²⁺ extracellular space. Free Calbryte^TM−520L was iontophoretically delivered by passing a constant current (−200 nA) throughout the experiment. Upon identifying the axon by negative staining (Fig. 4b–d), the EF was ceased, and the fluorescence around the axon decreased. Then, a test EF was applied, and the fluorescence increased on both sides (Fig. 4e, f). These increases seemed to be due to EF-moved Ca²⁺ because the fluorescence decreased when the direction of test EF was reversed (Supplementary Fig. 8a, b, $n = 5$ axons). The integral value of the increase in the fluorescence intensity during the forward test EF was larger on the anodal side than on the cathodal side; the mean of anodal/cathodal ratios was 111.7 ± 1.0% (mean ± s.e.m., $n = 12$ axons from four independent experiments, two-tailed *t*-test, $P < 0.001$, Fig. 4g). These ratios exceeded the mean of anodal/cathodal ratios of the fluorescence intensities before EF application (106.4 ± 0.6%, mean ± s.e.m., $n = 12$, Fig. 4g, blue horizontal line, cf. Supplementary Fig. 8c). These results may suggest that the anodal surface of an axon encounters EF-moved Ca²⁺ more frequently than the cathodal surface in the directional movement

of Ca²⁺. The encounter of Ca²⁺ would be more asymmetric in the confined in vivo ECM[28].

**Asymmetric microtubule stabilization.** If the directionally moved Ca²⁺ regulates integrin asymmetrically, it would result in asymmetric reorganization of the cytoskeleton, including actin filaments and microtubules. In the present study, RGC axons did not extend out from the retinal strip in the presence of cytochalasin D, an inhibitor of actin polymerization (two retinae tested, Supplementary Fig. 9). However, axons without filopodia cultured with cytochalasin D can exhibit galvanotropic orienting behavior, although they grow slower[29]. This raised the possibility that electric axon steering depends at least partly on microtubule dynamics.

Microtubules take two states in a growing axon: dynamically unstable microtubules and stable microtubules[30–32]. Stable microtubules change into unstable microtubules upon the phosphorylation of MAP1B by GSK3β[30]. Since GSK3β and MAP1B have been shown to have functional roles in growth cone steering[33,34], an inhibitor of GSK3β, LY2090314, was tested. LY2090314 at 2.5 nM enhanced the ventral extension without EF [Fig. 5a, i, cf. Figure 1k (no), two-tailed *t*-test, $P < 0.001$]. No further increase was observed with EF (Fig. 5b, i). LY2090314 at 10 nM still increased the basal extension [Fig. 5c, i, cf. Figure 1k (no), two-tailed *t*-test, $P < 0.001$] with no effect of EF (Fig. 5d, i). However, LY2090314 at 50 nM decreased the basal extension [Fig. 5e, i, cf. Figure 1k (no), two-tailed *t*-test, $P < 0.001$] with no effect of EF (Fig. 5f, i). The enhancement and suppression of the ventral extension without EF could be explained by two roles for microtubules in axon extension. Dynamic microtubules interact with F-actin at the leading edge to promote growth cone advance[31]. The high dose of LY2090314 would have caused a

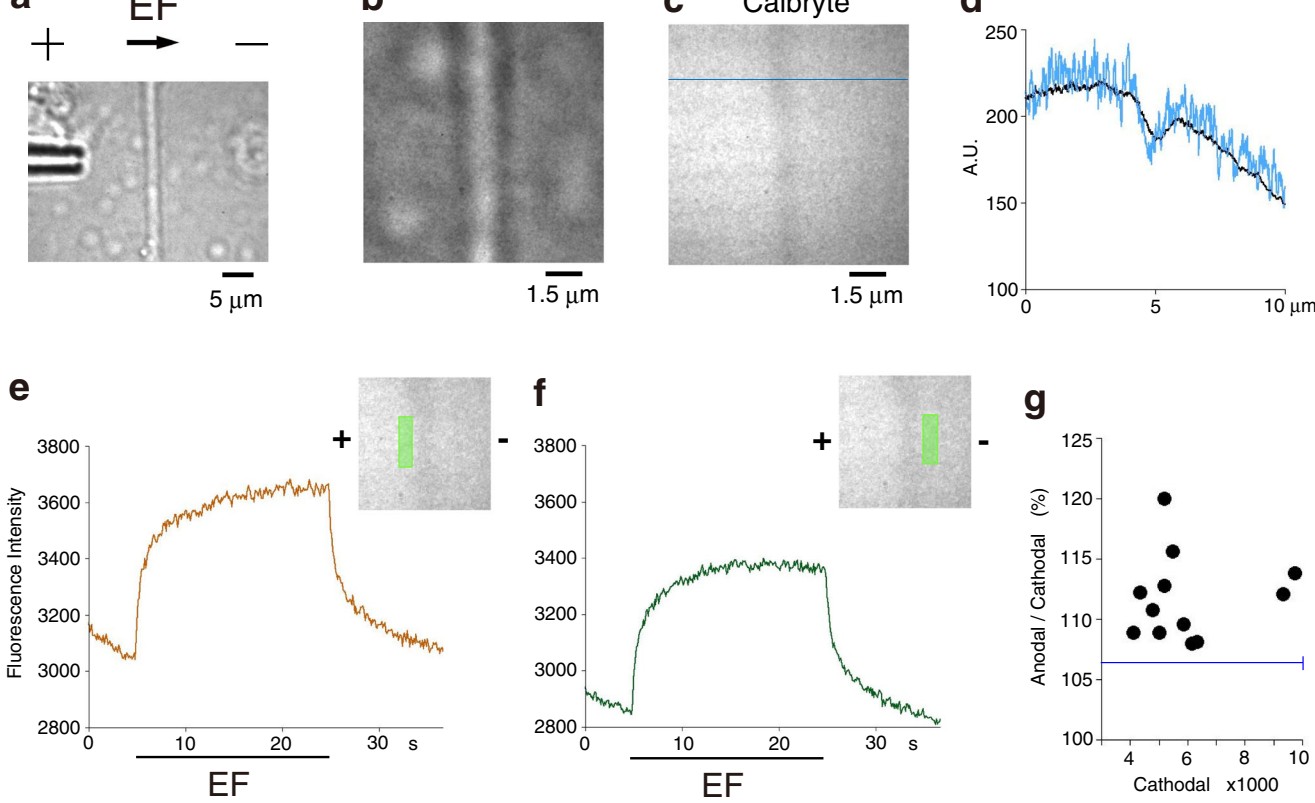

**Fig. 4 Ca²⁺ fluorescence recordings from the anodal and cathodal sides of an RGC axon. a** A transmission image of an RGC axon cultured in Matrigel®-based thin layer dissociated culture. A micropipette containing 1 mM CaCl₂ and 10 mM Calbryte™−520L was positioned at the anodal side of the axon. **b** A magnified image of the axon in (**a**). **c** A negative staining image of the RGC axon in (**b**). Calbryte™−520L diffused into the extracellular space. The horizontal stripe pattern was due to the rotation of Nipkow disk in the confocal scanner. **d** The transverse profile of a line scan at the horizontal blue line in (**c**) (blue) and the transverse profile of the whole fluorescence image of (**c**) (black) represented with mean fluorescence intensity/pixel. **e**, **f** Ca²⁺ fluorescence recordings from the anodal (**e**) and cathodal (**f**) sides of the axon in (**c**). Insets show each recording area in light green in the same scale as in (**c**). The fluorescence intensity is presented with 64-bit depth. **g** Anodal/cathodal ratio of integral values of increases in fluorescence intensity during EF (fluorescence intensity × time) plotted against the cathodal values measured on twelve RGC axons. The horizontal blue line indicates the mean anodal/cathodal ratio of the fluorescence intensities before EF application (vertical line represents ± s.e.m.). The axons measured were perpendicular to the direction of EF (90.6° ± 1.6°, mean ± s.e.m., $n = 12$ from 4 experiments).

lack of dynamic microtubules. On the other hand, the low dose of LY2090314 would have upregulated the stabilization of growing microtubules, preserving the dynamic microtubules for growth cone advance. Growing microtubules consist of dynamic unstable microtubules that elongate from the plus end of a stable microtubule[32]. Dephosphorylated MAP1B binds to these growing microtubules to stabilize them[31], which elongates the axon shaft. Next, SC79, an activator of Akt, was tested since Akt (PKB) inhibits GSK3β[35] and is activated downstream of integrin activation through the activation of phosphatidylinositol-3 kinase (PI3K)[18,36], of which inhibition enhanced the ventral extension without EF and abolished EF effects (Supplementary Fig. 10). SC79 at 25 µM also enhanced the ventral extension without EF [Fig. 5g, i cf. Figure 1k (no), two-tailed *t*-test, $P < 0.001$]. The EF did not increase the ventral extension any further (Fig. 5h, i). Figure 5j summarizes the lack of EF effects in the presence of LY2090314 and SC79. The loss of electric effects may suggest that the inhibition of GSK3β is essential for electric axon steering.

Assuming that integrin activation leads to GSK3β inhibition through Akt activation to stabilize microtubules, asymmetrically activated integrin would stabilize microtubules asymmetrically. Therefore, the distribution of microtubules was examined in the axons turning toward the cathode by staining with TubulinTracker™ Green (Fig. 5k, l). The asymmetry index (AI,

1.00 means symmetry) was obtained from the fluorescence image (Fig. 5m, Supplementary Fig. 11a, b). The AI indicated that microtubules were asymmetrically situated more on the inner (i.e., cathodal) side at the turning point; the mean of AIs was 2.02 ± 0.17 (mean ± s.e.m., $n = 6$ axons from five independent experiments, two-tailed *t*-test, $P < 0.01$, Supplementary Fig. 11a, c–g). These inner microtubules were stable because they existed after 30 min of nocodazole treatment, within which dynamic microtubules disappeared[32] (Supplementary Fig. 11g). F-actin did not show such a one-sided distribution (Fig. 5n, o).

## Discussion
Integrin and its ECM ligand are essential for the correct guidance of RGC axons in vivo[12]. However, it remains unknown how the integrin-ligand interaction steers the axon toward the future optic disc. Integrin-based adhesions underpin growth cone motility[37]. Integrin-mediated point contacts are asymmetrically distributed more on the side to extend in a turning growth cone[38,39]. Integrin-mediated adhesions underlie protrusions of growth cones to an attractive cue[40]. Axon growth rates correlate with the formation of integrin-based adhesions in response to positive and negative cues[41,42]. Thus, integrin is a crucial player in the regulation of axon growth and steering.

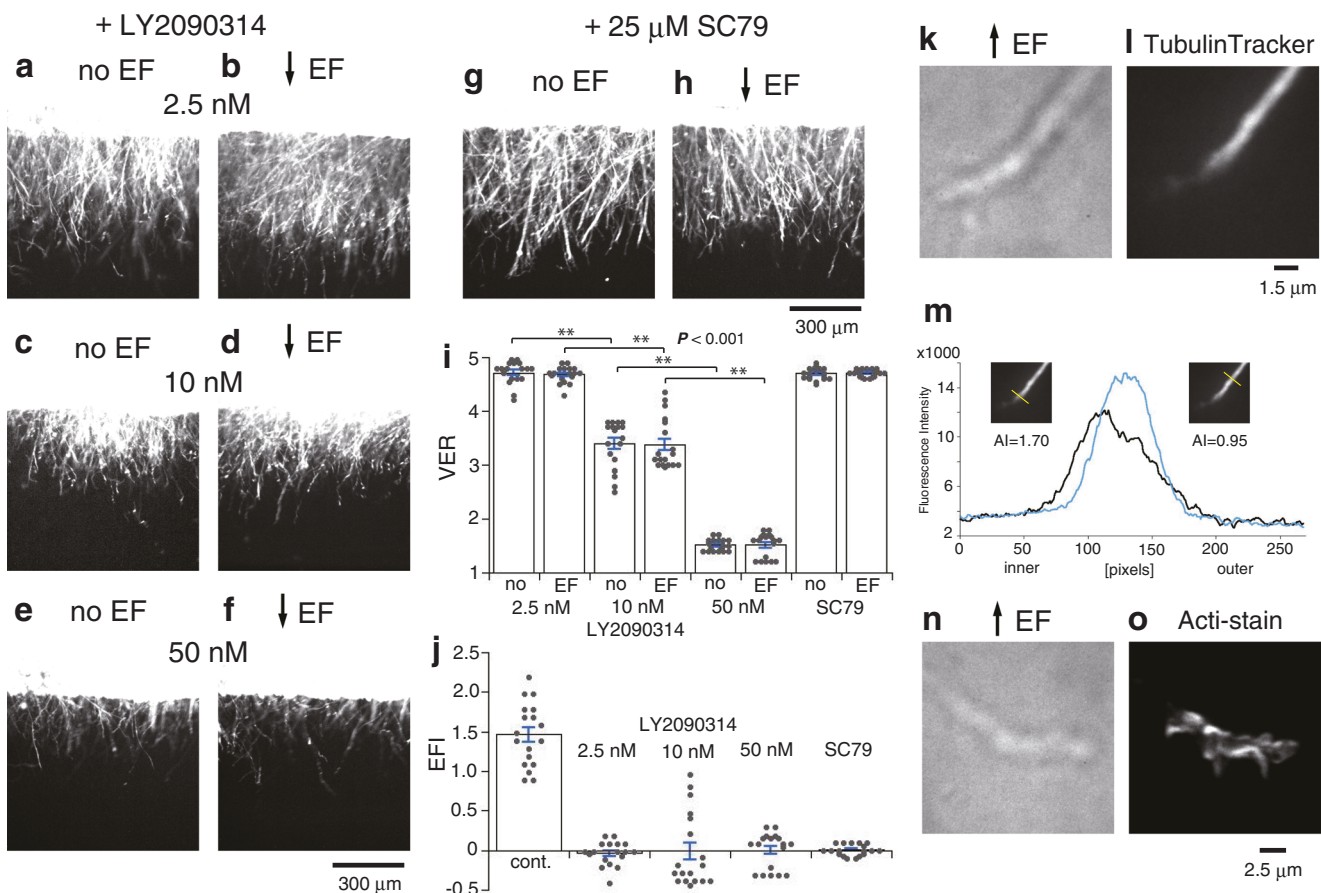

**Fig. 5 Effects of GSK3β inhibition and Akt activation and asymmetric microtubule distribution. a** RGC axons cultured with 2.5 nM LY2090314. **b** RGC axons cultured with 2.5 nM LY2090314 in the ventrally directed EF. **c** RGC axons cultured with 10 nM LY2090314. **d** RGC axons cultured with 10 nM LY2090314 in the ventrally directed EF. **e** RGC axons cultured with 50 nM LY2090314. **f** RGC axons cultured with 50 nM LY2090314 in the ventrally directed EF. **g** RGC axons cultured with 25 μM SC79. **h** RGC axons cultured with 25 μM SC79 in the ventrally directed EF. **i** VERs (mean ± s.e.m.) of retinal strips cultured with 2.5 nM, 10 nM, and 50 nM LY2090314, and 25 μM SC79 without EF (no) and with the ventrally directed EF (EF). Each column represents the value obtained from 18 photos taken at different focus levels from 3 retinal strips. Horizontal bars and asterisks denote significant differences (two-tailed $t$-test, $**P < 0.001$). **j** EFIs (mean ± s.e.m.) without drug (cont.) and with LY2090314 and SC79. The control was replotted from Supplementary Fig. 1e (cont.). **k**, **l**, **n**, **o** RGC axons turning dorsally in the dorsally directed EF. They extended out from the ventral edge of retinal strips. The EF was applied for 3 h before fluorescence staining. **k** A transmission image of an RGC axon. **l** A fluorescence image of the axon in (**k**) stained with TubulinTracker$^{TM}$ Green. **m** Transverse profiles of the fluorescence image in (**l**) at the turning point (left, black) and the straight part (right, blue). Insets show each scanning line in yellow with the asymmetry index (AI, see Supplementary Fig. 11a, b). One pixel size: 14.4 × 14.4 nm. **n** A transmission image of an RGC axon. **o** A fluorescence image of the axon in (**n**) stained with Acti-stain$^{TM}$ 488 fluorescent phalloidin.

The results of Ca$^{2+}$ fluorescence measurements suggested that the anodal surface of an axon encounters EF-moved Ca$^{2+}$ more frequently than the cathodal surface by at least 10%. Since the integrin headpiece containing ADMIDAS occupies ~1 zL ($10^{-21}$ L)[43] and the density of Ca$^{2+}$ is ~0.6/zL in 1 mM Ca$^{2+}$, the directionally moved Ca$^{2+}$ would bind to ADMIDAS asymmetrically. As Ca$^{2+}$ binding to ADMIDAS inhibits integrin-ligand binding[11], asymmetric Ca$^{2+}$ binding would result in less inhibition of integrin on the cathodal side. The ligand-bound integrin forms clusters to send intracellular signals for integrin-cytoskeleton linkage[36,44]. The cluster formation may be facilitated by a positive feedback mechanism[45], which could amplify the electric effect.

The asymmetrically activated integrin would activate PI3K-Akt pathway asymmetrically. The asymmetric PI(3,4,5)P$_3$ and Akt signaling has been demonstrated to serve as an early regulator during attractive turning of a growth cone[46]. As Akt (PKB) inhibits GSK3β[35], the less inhibited integrin would inhibit GSK3β more on the cathodal side. Since GSK3β decreases stable microtubules to maintain dynamic microtubules through MAP1B

phosphorylation[30], the more inhibition of GSK3β would increase stable microtubules on the cathodal side. Thus, the less inhibition of integrin would lead to more stabilization of microtubules. In a turning axon, microtubules are stabilized on the side to turn in the growth cone and growing shaft[30,47]. The asymmetric microtubule stabilization would steer the axon toward the cathode (Supplementary Fig. 12). The microtubule fluorescence showed that the peak fluorescence was cathodally shifted at the half level of fluorescence intensity (Fig. 5m and Supplementary Fig. 11). The red vertical line in Supplementary Fig. 11 indicates the center of the half fluorescence level, not the exact axon center. However, since the length of the half fluorescence level is ~100 pixels and one pixel size is 14.4 nm, the half fluorescence level almost covers the diameter of an axon. An asymmetric fluorescence distribution was found at a turning point (Supplementary Fig. 11a), not found at straight parts (Supplementary Fig. 11b). Thus, microtubules are likely to be asymmetrically distributed at a turning point of an axon in response to an EF. Other substrates for GSK3β, APC and CRMP2 promote microtubule capture and assembly upon the inhibition of GSK3β[48]. APC and CRMP2 could also contribute to

cathodal steering upon asymmetric GSK3β inhibition. GSK3β also acts as a key regulator in EF-induced directional migration of neural progenitor cells[49] and microglia[50].

The difference in the dose–response relationships between TASC and W1B10 (Fig. 1l) may be due to their epitopes; TASC binds to the region near the transmembrane domain[51], whereas W1B10 is considered to bind to the ligand-binding domain[52]. It could be postulated that TASC facilitated cluster formation upon expression of its epitope in the ligand-bound open conformation, which could promote adhesion to laminin[25]. On the other hand, W1B10 may have facilitated dissociation of integrin from the ligand[26]. The dissociation may be necessary for and promote axon extension because high $Mn^{2+}$ suppressed axon extension (Supplementary Fig. 2) and because $Mn^{2+}$ abrogated the enhancement of axon extension by W1B10 (Fig. 2g). The dissociation and inactivation of integrin would maintain the dynamic microtubules for growth cone advance. This idea could be supported by the fact that the inhibition of PI3K enhanced the basal extension of RGC axons (Supplementary Fig. 10). In summary, an axon is extended by two ways: (1) axon shaft elongation by stable microtubules and (2) growth cone advance by dynamic microtubules. TASC would have enhanced the former process by increasing stable microtubules through GSK3β inhibition, whereas W1B10 would have promoted the latter process by providing dynamic microtubules.

The central nervous system (CNS) including the retina develops as neuroepithelial cells proliferate, during which the first-born neurons extend long-distance traveling axons, such as commissural axons and optic nerves[53,54]. The mechanisms for the correct orientation of these long, first axons have been unknown. The bipolar-shaped neuroepithelial cells generate positive potentials in the extracellular space in the basal region of neuroepithelium through their apico-basal sodium transport[4,7,8]. As these cells most actively proliferate, increasing cell density at the dorsal portion of the neural tube[55], the most positive potential is likely to be established at the dorsal part of the neuroepithelium[4]. In fact, the highest potential is recorded at the dorsal part of the retinal neuroepithelium[7], where the cells most actively proliferate[56]. On the other hand, the cells located at the ventral portion of the optic cup are triangular-shaped[3], similar to the wedge-shaped floor plate cell in the neural tube[57]. This morphology implies that these ventral cells are not involved in apico-basal sodium transport. The ion channels of the floor plate cell are not responsible for sodium transport[58]. Thus, the positive potential is not generated at the ventral part of the developing CNS, including the retina[7]. Therefore, the resulting EF is oriented in the dorsoventral direction. This EF would direct the first axons, as demonstrated in the retina[7]. The guidance cues for commissural axons are still enigmatic; a question (-?-) is raised in Extended Data Fig. 4h of Dominici et al.[2], where "+" indicates netrin, "−" indicates a putative repellent molecule in the dorsal region[59–62]. The dorsoventral EF may also direct commissural axons toward the floor plate. As neuroepithelial cells generate EFs during neurogenesis, galvanotropism may primarily be used in CNS development.

## Methods

**Retinal strip culture.** Fertilized eggs were purchased from a local farmer (Asian Rural Institute, Nasushiobara, Japan). The study protocol was approved by the animal ethics committee of International University of Health and Welfare. A neural retina isolated from a chick embryo incubated for six days (embryonic day 6, E6) was spread on a black membrane filter (13006-025N, Sartorius) with the inner side up. The retina-membrane assembly was cut into a strip in the naso-temporal direction at the segment 0.7 mm dorsal to the optic nerve head (Fig. 1a). The size of the retinal strip was 0.8 mm in width, 2.4–2.7 mm in length (Fig. 1b). To identify the original direction of the retinal strip, the temporal end was cut perpendicularly to the dorsal and ventral edges, and the nasal end was cut obliquely

to them with the ventral edge longer. A wedge-shaped membrane filter was placed between the nasal edge and the wall of the trough of the culture chamber to fix the retinal strip (Fig. 1b). It was absolutely necessary that the retinal strip was completely flat-mounted. If the retinal strip was detached from the membrane filter at the periphery during incubation, retinal folding disturbed the direction of outgrowing axons. Shrinkage or retraction of retinal strip from the edge of membrane filter resulted in a non-straight line of the ventral edge of retinal strip. These retinae were excluded from the data. Protrusion of cell mass from the surface of retinal strip caused diverse, irregular, long extensions of axons. These retinae were also excluded. The complete protocol for retinal strip culture is as follows[63].

*Preparation of culture medium and culture chamber.* DMEM (041-29775, low glucose with L-glutamine and phenol red, FUJIFILM Wako, Osaka, Japan) was added to 3.6 mL of a serum mixture [25 mL FBS (Biosera) and 5 mL chicken serum (Gibco®), stored at 4 °C] in a 50 mL conical tube so that the total volume became 30 mL (final serum concentrations: FBS 10%, chicken serum 2%). The serum mixture was put on ice. The conical tube was put in a dry incubator at 38 °C. Both walls of the trough (22.0 mm in length, 3.2 mm in width, and 2.0 mm in depth) of the culture chamber (Φ25, 2 mm in thickness, made from an acrylic disc) were coated with silicone grease (Dow Corning, H.V.G) by using a fine rod (a wooden toothpick), which was cleaned with Kimwipes® and ethanol before use. Without the coating, Matrigel® (356234, Corning®) adhered to the wall of the trough. A cover glass (Φ25) was securely attached to either side of the culture chamber with silicone grease to make the bottom of the trough. A small amount of silicone grease was put on the upper surface of the culture chamber along both walls of the trough to securely attach another cover glass (Φ15) afterward.

*Preparation of Matrigel® and cooling.* An aliquot of Matrigel® (152 µL, stored at −20 °C) was put in a refrigerator at 4 °C. The aliquot of Matrigel® was put in ice after 30 min. A serum mixture of 48 µL was added to the Matrigel® (final serum concentrations: FBS 20%, chicken serum 4%). If a drug was applied, the drug was added to the Matrigel®. The Matrigel® containing sera (and the drug) was cooled in ice and homogenized by tapping. The cooling and tapping were repeated at least three times. The Matrigel® and an aluminum block (Φ25, 15 mm in thickness) were put in ice. A 35 mm dish was put on ice and filled with DMEM for cooling of a 10 µL pipette tip by sucking the ice-cooled DMEM before sucking Matrigel®.

*Isolation of retina.* Eggs incubated for six days were brought to a clean room. The egg was put on an egg stand with the air chamber up. The shell was broken and the embryo was exposed in a petri dish. The neck was cut with a forceps and the head was scooped and transferred to a 35 mm dish filled with a Hanks' balanced salt solution (HBSS) modified without $Ca^{2+}$, $Mg^{2+}$ (H6648, Sigma-Aldrich) by using a ring (Φ6-7) of stainless wire, which was made by twisting a thin stainless wire and connected to a holder. Several heads were compared for staging. An eye was enucleated under a dissecting microscope and transferred to another 35 mm dish filled with HBSS by holding the lens with a fine forceps. Extraocular muscles were removed with a pair of fine forceps. A small cut was made along the equator near the optic fissure. The eye was transferred to another 35 mm dish filled with HBSS containing a black membrane filter (13006-025N, Sartorius). The pigment epithelium was peeled away with a pair of fine forceps by inserting the tip of forceps into the cleft between the neural retina and the pigment epithelium to separate them. The optic nerve head was cut with a fine forceps. The optic nerve and the pigment epithelium were removed. A continuous cut was made along the equator to separate the anterior retina from the posterior retina. The eye was put on the black membrane filter with the lens up. The eye was gently pushed onto the membrane filter so that the outer limiting membrane of the posterior retina was attached to the membrane filter. The vitreous body was separated from the posterior retina. The posterior retina was attached to the membrane filter by pressing the periphery on the membrane filter. The vitreous body could be separated from the retina at the dorsal, nasal and temporal parts. Then, the vitreous body was separated from the optic fissure. The whole vitreous body and the anterior retina were removed together with the lens. The whole posterior retina was attached to the membrane filter by pressing the periphery on the membrane filter with a fine forceps (Fig. 1a).

*Preparation of retinal strip.* The retina-membrane assembly was lifted up from HBSS and put on a filter paper (Whatman®, Φ55, 1001-055) with the retina side up. The filter paper was put under the dissecting microscope to position the optic fissure vertically. A 15 mm stainless scale with the minimum scale of 0.5 mm was put near the retina on the left in parallel with the optic fissure. The level of the optic nerve head was defined by seeing the scale. A horizontal cut was made at the level of the optic nerve head in the nasotemporal direction (i.e., perpendicularly to the optic fissure) with a piece of razor blade (0.1 mm in thickness, 8 mm in width) attached to a blade holder. By this cutting the level of the optic nerve head was defined on the scale. A horizontal cut was made 0.7 mm dorsal to the previous cut in parallel with it. This cut made the ventral edge of the retinal strip. Another horizontal cut was made 0.8 mm dorsal to the previous cut in parallel with it. This cut made the dorsal edge of the retinal strip defining the width of the retinal strip. A vertical cut was made 1.2 mm left to the optic fissure. An oblique cut was made 1.2–1.5 mm right to the optic fissure with the ventral edge longer. This oblique cut

defined the direction of the retinal strip (Fig. 1b). A piece of parafilm was attached to the bottom of a 35 mm dish. A drop of DMEM (25 μL) was made on the parafilm. If a drug was applied, the drug was added to the DMEM for pre-incubation. The retinal strip was transferred into the drop of DMEM by holding the lower right corner of the retinal strip with a fine forceps.

*Culture of retinal strip.* The ice-cooled aluminum block was put in a petri dish. The space around the aluminum block was filled with ice. The culture chamber was put on the aluminum block. One microliter of the ice-cooled Matrigel® containing sera was dropped on the bottom of the trough of the culture chamber. The bottom of the trough was swept with a fine tungsten rod, which was polished electrolytically, so that the bottom was coated with the Matrigel®. A wedge of membrane filter was put on the bottom of the trough. This wedge was used to fix the retinal strip (Fig. 1b). The retina-membrane assembly was put on the bottom of the trough with the retina up. The retinal strip was positioned perpendicularly to the wall of the trough using the fine tungsten rod. The retina-membrane assembly was attached to the bottom by reducing the space between the retina-membrane assembly and the bottom to prevent it from floating. The temporal edge of the retinal strip was attached to the wall on either side of the trough at the middle of it. The retinal strip was fixed in the trough by the wedge of membrane filter (Fig. 1b). Ten microliters of the ice-cooled Matrigel® containing sera was sucked and 8 μL was dropped over the retinal strip by leaving 2 μL in the 10 μL pipette tip not to add air bubbles. If air bubbles were seen in the Matrigel®, they were removed with the fine tungsten rod. A cover glass (Φ15) was put over the retinal strip and it was securely attached to the upper surface of the culture chamber with silicone grease. The culture chamber was put in a humidified thermostat chamber at 37 °C for gelling. Gelling could also be done before dropping 8 μL of Matrigel® to prevent detachment of retinal strip from membrane filter. After five minutes of gelling, the culture chamber was put on a yellow paper (Φ25), which was placed on an acrylic disc (Φ30, 10 mm in thickness) for easy separation of the culture chamber from the acrylic disc after incubation. The yellow papers between black membrane filters stored in a plastic case were used for the separation. The 50 mL conical tube containing DMEM with 10% FBS and 2% chicken serum was taken out from the dry incubator. The trough of the culture chamber was filled with the DMEM (about 100 μL) by using a fire-polished glass Pasteur pipette. The DMEM was carefully injected from either end of the trough to avoid air bubbles. The acrylic disc with the culture chamber was put in a corner well of a 6-well culture plate. The neighboring two wells were filled with an excess volume (14 mL) of DMEM containing 10% FBS and 2% chicken serum to prevent changes in the composition of the medium around the cell due to electrophoresis. The both ends of the trough were bridged to the neighboring wells filled with the DMEM by using a pair of U-shaped glass tubes (Φ3) completely filled with the culture medium to avoid air bubbles. The two wells filled with the culture medium were connected by using another pair of U-shaped glass tubes (Φ3) filled with 1% agar-saline to the neighboring two wells, which were filled with DMEM buffered with 25 mM HEPES (048-30275, high glucose with L-glutamine and phenol red, FUJIFILM Wako). Thus, five wells were used for a current flow. The remaining one well was filled with distilled water to maintain humidity. Distilled water (1 mL) was injected into the well containing the culture chamber around the acrylic disc. The 6-well culture plate was put in an airtight jar (2.5 L in volume). A pack of AnaeroPack®·CO₂ (Mitsubishi Gas Chemical Company, Inc., Tokyo) was put in the airtight jar. AnaeroPack®·CO₂ produces CO₂ and maintains 5% CO₂ in an airtight jar for at least 150 h. The lid of the airtight jar was closed. The airtight jar was put in a dry incubator at 38 °C and incubated for 24 h.

**Constant EF culture**. The culture system to provide a constant EF contained the anode and cathode electrodes for current supply and a pair of glass micropipette electrodes to record voltages in the culture medium. The amount of current supplied to the culture medium was regulated by a negative feedback circuit to maintain the voltage difference between the two points (15 mm in distance) along the current flow at the desired value (225 mV for 15 mV/mm). The complete protocol for constant EF culture is as follows[64].

*Electrodes for supplying current and monitoring voltages.* The anode and cathode Ag/AgCl disc electrodes (Φ15−25) were placed into the two wells filled with HEPES-buffered DMEM. The electrodes for monitoring voltages in the culture medium were made from a Φ3 glass tube by heating it over a small fire and pulling it until the tip diameter became 0.3 mm. The tip and the cut edge of the glass pipette were fire-polished. The glass pipette electrodes were filled with 1% agar-saline by using a silicone tube (inner Φ2.5) and a 2.5 mL syringe. A small Ag/AgCl pellet (Φ1) containing an Ag/AgCl wire was inserted into the glass pipette electrodes. The Ag/AgCl wire was soldered to a copper lead wire. This junction was covered with epoxy resin to prevent contact with the agar-salt gel. The lead wire was connected to the input of a voltage follower. An LMC662CN (Texas Instruments) operational amplifier was used for the voltage follower because the input resistance was more than 1 TeraΩ and the bias current was 2 fA. To minimize the length of the lead wire connected to the monitor electrode and the voltage follower, the LMC662CN was placed near the culture chamber in the airtight jar. The cables to the anode and cathode electrodes, the outputs of the two voltage followers, and the power supply to the LMC662CN were connected to the connectors fixed onto the wall of the airtight jar with silicone grease. The monitor electrodes were

prepared just before the preparation of retinal strip. The monitor electrodes were cleaned in boiling water after each use.

*Electrical circuit with negative feedback.* The two monitor electrodes were inserted into the culture medium in the trough at the both edges of the top cover glass (Φ15) to keep the distance between the two monitor electrodes at 15 mm. An electrodes holder was made from an acrylic box with two holes (Φ3.2). The voltage difference between the two points was continuously monitored using a differential amplifier. The output of the differential amplifier was connected to the inverting input (−) of an operational amplifier (BB3582J, Burr-Brown). BB3582J was used because of its wide operating voltage range (±115 V). A reference voltage (Vref) was used to regulate the voltage difference between the two points by the negative feedback circuit. Vref was connected to the non-inverting input (+) of BB3582J. The feedback current flowed through the culture medium between the two monitor electrodes, the total resistances of the anode and cathode electrodes, the agar-salt bridges, the glass tubes containing the culture medium, and the solutions connected to them, and the feedback resistor (Rf). Rf was adjusted by using a variable resistor (0.1–1.0 MΩ). The output of the differential amplifier equaled Vref upon the negative feedback by the operational amplifier. The offset voltage of the differential amplifier, including offsets of the two monitor electrodes and the two voltage followers, was nullified by subtracting the output voltage recorded during current-off periods of 0.5 s at an interval of 100 s (duty cycle, 99.5%). The current flow was interrupted with a pulse-driven relay switch. The output of the differential amplifier was continuously recorded with a chart recorder (PowerLab 2/26, ADInstruments) throughout the entire incubation period to check that all components were operating properly. Since the current amplitude to maintain an EF strength of 15 mV/mm was around 180 μA (150–200 μA), the specific resistance of the culture medium was estimated to be ~53 Ωcm.

**Solutions**. A Hanks' balanced salt solution (HBSS) modified without $Ca^{2+}$, $Mg^{2+}$ (H6648, Sigma-Aldrich) was used for preparation of retinal strip. A retinal strip was embedded in 8 μL of Matrigel® (356234, Corning®) containing 20% FBS (Biosera) and 4% chicken serum (Gibco®). The trough of the culture chamber (100 μL in volume) was filled with DMEM (041-29775, low glucose with L-glutamine and phenol red, FUJIFILM Wako, Osaka, Japan) containing 10% FBS and 2% chicken serum. Both ends of the trough were connected to the neighboring wells filled with the same culture medium of excess volume (14 mL) by using a pair of glass tubes filled with the culture medium. Penicillin-Streptomycin-Amphotericin B suspension (161-23181, FUJIFILM Wako) was added to the culture medium at 1%. The anode and cathode electrodes were placed in the other wells filled with DMEM buffered with 25 mM HEPES (048-30275, high glucose with L-glutamine and phenol red, FUJIFILM Wako).

**Fluorescence staining and imaging**. Live cells and axons were stained with calcein-AM (10 μM, DOJINDO) in DMEM (044-32955, high glucose with L-glutamine, HEPES buffered, without phenol red, FUJIFILM Wako) for 30 min at room temperature. A Nipkow-type confocal scanner (CSU10, Yokogawa, Kanazawa, Japan), an sCMOS camera (ORCA®-Flash4.0 V2, Hamamatsu photonics, Hamamatsu, Japan), and HSR software (Hamamatsu) were used for fluorescence imaging. Ventral extension rate (VER) was obtained by the ratio of the mean fluorescence intensity between the lines 150 μm and 400 μm ventral to the retinal strip along the whole ventral edge against the background intensity measured at the area without the retina (Fig. 1c, d). Photos for VER measurement were taken with a ×4 objective at the level of RGC layer, which provided representative images. ImageJ was used to measure mean fluorescence intensities in a range of 0–255. Background intensity was adjusted to around 21 by using Photoshop® Elements.

**Antibodies and drugs**. TASC (MAB19294) and W1B10 (I8638) were purchased from Millipore and Sigma, respectively. A monoclonal mouse IgG1 isotype control antibody M075-3M2 (Functional Grade) was purchased from MBL (Nagoya, Japan). TASC was applied at 20, 50, 100, 200 μg/mL. W1B10 was applied at 10, 20, 50, 100, 200 μg/mL. M075-3M2 was applied at 100, 200 μg/mL. Retinal strips were preincubated in a drop of DMEM (25 μL) containing the antibody on Parafilm® for 10 min. Since the antibodies were presented in phosphate buffer, decreases in free $Ca^{2+}$ and $Mg^{2+}$ due to the phosphate buffer were compensated by adding $CaCl_2$ and $MgCl_2$ to the preincubation medium and Matrigel® at the ratio of 4 (phosphate buffer): 2 ($CaCl_2$): 1 ($MgCl_2$). LY2090314 (Selleck) was dissolved in DMSO at 10 mM as a stock solution and diluted to 1 mM for an aliquot of 25 μL. SC79 (Selleck) was dissolved in DMSO at 25 mM as a stock solution. Omipalisib (GSK2126458, GSK458, Selleck) was dissolved in DMSO at 5 mM as a stock solution. Cytochalasin D (Cayman Chemical) was dissolved in DMSO at 1 mg/mL as a stock solution. Calbryte$^{TM}$−520L potassium salt (20642, AAT Bioquest®) was dissolved in distilled water at 10 mM containing 1 mM $CaCl_2$. Nocodazole (FUJIFILM Wako) was dissolved in DMSO at 10 mg/mL as a stock solution.

**Organotypic culture**. Optic cups were isolated from E4 chicks. The lens and the pigment epithelium were eliminated. The whole neural retina was incubated for 24 h in DMEM containing 20% FBS and 4% chicken serum. After the incubation,

the retina was spread on a black membrane filter with the inner side up or on the bottom of a recording chamber with a pair of L-shaped tungsten needles.

**Focused EF culture with fan-shaped microfluidic chip.** Retinal strips from E6 chicks were cultured for 48 h. The ventral side of the retinal strip faced the open side of a fan-shaped microfluidic chip (Supplementary Fig. 3, SOKEN, Ome, Japan). It was made from an acrylic plate (3.0 mm in width, 2.0 mm in length, 0.8 mm in thickness). Its opening angle was 160°. The central channel (0.2 mm in width, 0.5 mm in depth) was dug in the middle for current flow. It was placed in the trough (3.2 mm in width) of a 1.0-mm-thick culture chamber. All gaps were filled with silicone grease so that the current flowed through the channel alone. A constant current was applied from a high-voltage amplifier of a constant output voltage (115 V) through a resistor (2, 20, 200 MΩ). Matrigel® and the culture medium contained 20% FBS and 4% chicken serum. W1B10 (100 μg/mL) was added to the preincubation medium and Matrigel® to enhance electric effects.

**Microchannel assay.** A culture chamber containing a microchannel was made from an acrylic disc (25.0 mm in diameter, 1.0 mm in thickness, SOKEN, Ome, Japan). The width of the microchannel was 0.2 mm, the length and the depth were 1.0 mm (Supplementary Fig. 5a). A current was focused on the microchannel in between the arcuate walls to flow to the cathode. Retinal strips from E6 chicks were cultured for 48 h. The length of the ventral edge of retinal strips was 2.2 mm and the width was 0.8–1.0 mm. The ventral edge of retinal strip faced the microchannel at a distance of 0.4 mm (Supplementary Fig. 5b). The nasal and temporal corners of the ventral edge of retinal strip made contact with the arcuate wall of the micro-channel chamber. A constant current was applied from a constant current source (GS200, Yokogawa). Matrigel® and the culture medium contained 20% FBS and 4% chicken serum. The convergence angle (CA) was measured by using ImageJ between two axons extending straight (≥100 μm) from the most nasal and temporal regions of retinal strip toward the microchannel. Sigmoid or curved axons were not used. Transverse profiles of fluorescence intensities at the half-distance line between the retina and the microchannel were made from original tif images by using ImageJ. The minimum value of fluorescence intensity was subtracted as background. $F_{CA}$ was obtained from the total sum of transverse profile. Photos for CA and $F_{CA}$ measurements were taken with a ×10 objective at the level of RGC layer.

**Thin layer dissociated culture.** Retinal strips (~2.5 mm × 2.5 mm) were dissected from the segment 0.7 mm dorsal to the optic nerve head of E6 chick retinae on a black membrane filter with the inner side up. The retinal strip was cooled in DMEM on ice and transferred to the culture chamber for constant EF culture. It was soaked in 2.5 μL Matrigel® containing 20% FBS and 4% chicken serum. The superficial layer of the retina was scratched out and spread on the bottom of the chamber with a fine painting brush, of which the hair bundle diameter was 0.5 mm (GH-BRSUP-GT, GodHand, Tsubame, Japan). The culture chamber was put in a humidified thermostat chamber at 37 °C for 5 min for gelling. Then, it was filled with DMEM containing 20% FBS and 4% chicken serum and incubated for 24 h.

**Ca²⁺ fluorescence recording.** The Matrigel® thin layer containing cell masses was washed five times with a Hanks' balanced salt solution modified without $Ca^{2+}$, $Mg^{2+}$ (H6648, Sigma-Aldrich). A micropipette was made from a borosilicate glass capillary containing a filament (GC150F-10, Harvard Apparatus) using a puller (P-97, Sutter Instrument). The micropipette was filled with 10 mM Calbryte™−520L (20642, AAT Bioquest®) containing 1 mM $CaCl_2$ and was manipulated by using a piezoelectric positioning system (PCS-1000, Burleigh Instruments). The current to deliver Calbryte™−520L (−200 nA) was supplied from a constant current source (GS200, Yokogawa). To apply an EF of ~15 mV/mm, a constant current of 180 μA was supplied from a high-voltage amplifier of a constant output voltage (115 V) through a variable resistor (0.1–1.0 MΩ) with a pair of polyethylene tube electrodes (Φ 1 mm) containing 1% agar-saline.

**High-magnification confocal microscopy.** The high-magnification confocal microscope system (Supplementary Fig. 13) was used for Ca²⁺ fluorescence recording and fluorescence imaging of cytoskeletons. It contained an image intensifier (C9016-01, Hamamatsu) and a high-power laser source (473 nm, 150 mW, 06-MLD, 0473-06-03-0150-100, Cobolt). Ca²⁺ fluorescence images were captured at a rate of 10 fps (exposure time: 100 ms). HCImage Live software (Hamamatsu) was used to capture and analyze Ca²⁺ fluorescence images. ImageJ was used to obtain transverse profiles and to measure the angle of axons.

**Microtubule staining.** TubulinTracker™ Green (Oregon Green™ 488 Taxol, bis-acetate, T34075, Invitrogen) reagent (Component A) was dissolved in 71 μL of DMSO to make a 1 mM stock solution. Ten microliters of the Component A was mixed with 10 μL of Component B (20% Pluronic F-127 in DMSO) to make a 500 μM intermediate stock solution. Ten microliters of the intermediate stock solution was dissolved in 10 mL DMEM (HEPES buffered, without phenol red) to make a 500 nM working solution. Retinal strips were stained with the working solution for 30 min at room temperature.

**Actin staining.** Acti-stain™ 488 fluorescent phalloidin (PHDG1, Cytoskeleton) was used. Fifty microliters of the reagent (14 μM) was diluted into 5 mL of phosphate-buffered saline (PBS, pH 7.4, Nacalai Tesque) to make a 140 nM working solution. Retinal strips were fixed in a fixative solution (4% paraf-ormaldehyde in PBS, Nacalai Tesque) for 10 min and permeabilized in a per-meabilization buffer (0.05–0.1% Triton X-100 in PBS) for 5 min. Then, they were stained with the working solution for 30 min. All the procedures were done at room temperature.

**Statistics and reproducibility.** One retinal strip was made from one retina. The number of retinal strips tested equals the number of independent experiments. Six photos were taken from one retinal strip and the total data from multiple retinal strips were averaged for each experimental condition. Two-tailed $t$-test was applied to all the statistical comparisons. Statistically significant differences were evaluated when $P < 0.001$ (**) or <0.01 (*). All the experimental findings presented in this study were replicable.

**Reporting summary.** Further information on research design is available in the Nature Portfolio Reporting Summary linked to this article.

## Data availability
The data that support the findings of this study are available from the corresponding author upon reasonable request. The numerical source data for Figs. 1d, k, l, 2g, 3d–f, 4d–g, 5i, j, m are provided in Supplementary Data 1.

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

## Acknowledgements

This work was supported by the Japan Spina Bifida and Hydrocephalus Research Foundation (JSBHRF), the Eye Research Foundation for the Aged (ERFA), Suzuken Memorial Foundation, the Naito Foundation, Novartis Alcon, JSPS KAKENHI JP25460298, JP18K06857, JP21K06772.

## Author contributions

M.Y. contributed to the design and implementation of the research, to the funding acquisition, to the analysis of the results and to the writing of the manuscript.

## Competing interests

The author declares no competing interests.
