## [Peer Review File · Communications Biology]

Reviewers' comments:

Reviewer #1 (Remarks to the Author):

In this paper titled "Electric axon guidance for optic nerve formation" the author examines how chick RGC axons respond to an electric field (EF). The author shows that the EF can direct growing axons towards the cathode in an integrin dependent manner through asymmetric microtubule stabilization. The experiments are straightforward but the methods by which they derive quantitative data are questionable. Also, the logical chain by which one set of experiments flows into the next does not necessarily make sense. The lack of data to explain why experiments were chosen when moving from one set of results to the next and the total absence of a substantial introduction/research background is unacceptable. This paper is unfinished and incoherent, and these issues should be fully addressed.

Major Comments:

1. As per the Communications Biology journal guidelines, a substantial introduction/research background are mandatory, and Abstract must not contain references. The lack of an introduction also causes issues in the results as the reader has no understanding as to what the rationale for moving from one set of results to the next makes logical sense.

2. The way the Ventral Extension Rate is calculated is not fully explained and draws questions as to the consistency of such quantitative results. The paper states that VER is calculated within the 150 μ m - 400 μ m range of the ventral retinal strip. Given that the explants cut from the retina may not be a straight line from left to right and that fig 1b shows that axon outgrowth along the ventral strip is not consistent, the results garnered from these results may be inconstant from one strip to the next. Additionally, the size of the strips cut from the retina are not specified, nor is it stated if the entire 150-400 μ m rectangle parallel to the strip is used to calculate the VER or if representative images were used for each strip. Regardless these need to be made clear in the materials and methods section at the bare minimum to avoid calling the results into question. Due to the potential variability and inability to track single axons in this experiment it may be beneficial to include an additional set of in vitro experiments tracking only a few neurons as they extend their axons in the presence of EF.

3. The reason for looking into integrin is not explained at all. Fully explaining the relationship between integrin and EF in the introduction would help bridge this jump in experimental decision making. However, this is not sufficient and better explanation as to why integrin was chosen to move from the first set of results to the second is necessary. It might also be helpful to show how loss of integrin impacts EF mediated axon outgrowth via knockdown experiments. This would help illustrate how integrin is the primary pathway influenced by EF.

4. The lack of a proper introduction also caused complications later in the results where the reader might make the incorrect assumption that Ca²⁺ dynamics shown in this paper concerning in influencing axon outgrowth might be novel. Regardless of if it was mentioned in the introduction, the rationale for pursuing experiments in asymmetric Ca²⁺ deposits in the axon should be explained. It should also be clarified in the discussion of this study's direct evidence and mechanisms by which EF affects Ca²⁺ axon outgrowth.

Minor Comments:

Line 7: "the existence of long-range chemoattractants was denied" does not make sense.

Line 8: Replace "that" with "and"

Line 22: The first section of the results needs a title

Line 160: delete "the" before "more"

Line 173: "The central nervous system (CNS) including retina" should have "the" before retina

Line 177: "the existence of long-range chemoattractants was denied" does not make sense.

Reviewer #2 (Remarks to the Author):

Summary

In the present work, Dr. Yamashita studied the contribution of steady electric fields (EFs) to the axon development of retinal ganglion cells and provides evidences supporting a role of B1 integrin in mediating axon response to EF.

Recent reviews testify of the renewed interest for bioelectric signals during neurodevelopment (ref 1–3). Nevertheless, EF contributions to axon guidance are largely overlooked. The paucity of examples in which EF manipulations could lead to quantified alterations of axon trajectories *in vivo* and the lack of knowledge of the molecular mechanisms setting response specificity impeding their acceptance as relevant cues.

Thus, providing a new mechanism of response to EFs and *ex vivo* demonstration of its contribution to axon guidance would be important.

In addition, although a significant part of the study relies on experimental paradigms mastered by the author (ref 4,5), novel interesting approaches are included in this paper (microfluidic, Ca^{++} imaging)

Nevertheless, I think that 3 aspects of the study could be improved to better support the author's conclusions. (1) The contribution of B1 integrin to the orientation of axon in response to EFs needs to be strengthened, (2) the role of B1 integrins as well as (3) other aspects of the mechanistic model need to be clarified. Each of the above concerns are detailed below.

Detailed comments:

1- The present results do not necessarily support that EFs guide the axons via B1 integrin regulation. On one hand, it seems that the quantifications of retina strip cultures better assay axon growth promotion in response to EF rather than EF control of axon orientation. Directly testing the ability of EF to control axon orientation would require a new set up where for example EF would be applied perpendicularly to the strip edge (as done in figure 3). On the other hand, the author did not quantify the experiments that relate more directly to EF-mediated control of axon orientation. Neither the effect of Mn^{2+} application on whole retina culture (Figure 2 h-k) nor the effect of EF on RGC convergence in microfluidic fan-shape (figure 2l-o and extended data figure 4) are quantified. Quantification of these experiments would be important to support the conclusions drawn. In addition, comparing the trajectory defects induced by the loss of EF (ref 4) to the ones due to Mn^{2+} addition (Figure 2 h-k) would help to support the contribution of B1 integrin to EF-mediated guidance in whole retina. The author needs to better evidence the contribution of EF to the axon convergence in the microfluidic device. As topography could guide axons, it might be useful to estimate the EF generated by the retina strip between its ventral edge and the funnel of the microfluidic device and/or to block transretinal potential with amiloride as previously done by the author (ref 4,5).

2- The author needs to clarify the effect of the different treatments used to manipulate integrin activity. The effect of the two antibodies used should be clearly presented and results should be interpreted accordingly. Indeed, TASC antibody was shown to increase adhesion on laminin while W1B10 inhibits adhesion on laminin (ref 6,7). It would be interesting to see which integrins pools are bound by the two antibodies in axons and growth cones. As basal outgrowth increases in presence of the antibodies, the author should clarify the net enhancement of EF-mediated ventral growth induced by the antibodies (Figure 1 and Extended data figure 1). Using another interfering tool such as the RGD peptide might be useful to clarify the integrin function.

The author needs to justify the Mn^{2+} concentration used as maximal integrin activation could occur for concentration under $100\mu M$ (ref 8) and that high concentration of Mn^{2+} are toxic for neurons (ref 9). Toxicity might explain why axon outgrowth is suppressed at high concentration (extended data figure 2). The potential side effects of Mn^{2+} may need to be considered when interpreting some of the results. As $100\mu M$ Mn^{2+} already decreases, by half, the hippocampal neurons viability (ref 9), one might consider that $500\mu M$ could have deleterious effects on RGC neurons. This might explain why Mn^{2+} -induced integrin activation did not increase axon length as the antibodies did (Figure 2a and g, Extended data figure 1), why large axon fascicles appear to be absent in whole retina culture (Figure 2 i, k) as well as different axon outgrowths within the retinal strip in the microfluidic device (figure 2 m and o). This suggests that RGC viability should

be checked or that lower concentrations should be tested.

3- Several improvements are required to support the elegant mechanistic model proposed by the author. It is unclear to me, how the author could ascertain the existence of an asymmetric concentration of Ca²⁺ at the surface of the axons. Calbryte -520L is described as cell-permeable and recommended to monitor intracellular calcium by some distributors (<https://www.aatbio.com/products/calbryte-520l-am>). Thus, the author should clarify this point and give the reference of the product. In addition, I think that measuring the fluorescence asymmetry with the Calbryte-containing micropipette on the cathode side could be an important control. This would ensure that the differences are not due to the micropipette position and rule out confounding factors such as subtle Calbryte gradient. Given the importance of growth cones in axon guidance and the specificity of microtubule dynamics in this structure, it seems to me that growth cones would have been more pertinent for this assay. Thus, author should explain the choice made. It would be important to discuss the anodal-cathodal difference of Ca²⁺ concentrations and its potential effects on integrin activation given the calcium affinities and effects of the different sites. Concerning the signaling downstream of integrin activation, it would be great to have a B1 integrin and GSK3 immunolabeling in the RGC growth cones. Higher resolution and quantification (asymmetric index) of the stabilized microtubule would be important to support the model.

From a formal point of view, I think that the addition of a proper introduction and a classical result section will help to appreciate better the questions and the context of the study. I recommend to the author to explain better the rationales for testing integrin function. Indeed, integrins were shown to be key component of the EF-oriented cell migration (ref 10–13) . I also think it would be helpful to better explain how integrins could regulate microtubules. It might be also important to consider the other signaling pathways (beside CRMP2 and APC) that GSK3 could regulate since some of them were shown to be important for galvanotaxis (ref 1,14,15). To better support the proposed model the author might also consider to discuss the role of integrin16–19 in growth cone response, the regulation of GSK3B activity in growth cones (ref 20) and the contribution of MAP1B to the turning of RGC growth cones (ref 21). Finally, although the discussion brings interesting ideas, I think that some of the statement should be less definitive. For example, the reference to the supplementary figure 4h of the paper from Dominici and collaborators, could lead readers to think that there are evidences supporting the contribution EF to commissural axon guidance while this figure only schematizes the distribution of netrin (symbolized with +) in the initial and new guidance model.

The author needs to indicate the p values for the different conditions compared (Figure1 j, extended data figure1d, Figure2g, Figure3g and Figure4i) and whether paired (figure3g) or unpaired t-test was used.

References:

1. Yu, X., Meng, X., Pei, Z., Wang, G., Liu, R., Qi, M., Zhou, J., and Wang, F. (2022). Physiological Electric Field: A Potential Construction Regulator of Human Brain Organoids. *Int. J. Mol. Sci.* 23, 3877. 10.3390/ijms23073877.
2. Levin, M. (2021). Bioelectric signaling: Reprogrammable circuits underlying embryogenesis, regeneration, and cancer. *Cell* 184, 1971–1989. 10.1016/j.cell.2021.02.034.
3. Medvedeva, V.P., and Pierani, A. (2020). How Do Electric Fields Coordinate Neuronal Migration and Maturation in the Developing Cortex? *Front. Cell Dev. Biol.* 8, 580657. 10.3389/fcell.2020.580657.
4. Yamashita, M. (2013). Electric axon guidance in embryonic retina: Galvanotropism revisited. *Biochem. Biophys. Res. Commun.* 431, 280–283. 10.1016/j.bbrc.2012.12.115.
5. Yamashita, M. (2016). Epithelial sodium channels (ENaC) produce extracellular positive DC potentials in the retinal neuroepithelium. *Data Brief* 6, 253–256. 10.1016/j.dib.2015.11.068.
6. Shih, D.-T., Edelman, J.M., Horwitz, A.E., Grunwald, G.B., and Buck, C.A. (1993). Structure/Function Analysis of the Integrin/31 Subunit by Epitope Mapping. *J. Cell Biol.* 122, 11.
7. Cruz, M.T., Dalgard, C.L., and Ignatius, M.J. (1997). Functional partitioning of $\beta 1$ integrins revealed by activating and inhibitory mAbs. *J. Cell Sci.*, 13.
8. Mould, A.P., Akiyama, S.K., and Humphries, M.J. (1995). Regulation of Integrin $\alpha 5\beta 1$ -

- Fibronectin Interactions by Divalent Cations. *J. Biol. Chem.* 270, 26270–26277. 10.1074/jbc.270.44.26270.
9. Daoust, A., Saoudi, Y., Brocard, J., Collomb, N., Batandier, C., Bisbal, M., Salomé, M., Andrieux, A., Bohic, S., and Barbier, E.L. (2014). Impact of manganese on primary hippocampal neurons from rodents: Impact of Manganese on Primary Hippocampal Neurons from Rodents. *Hippocampus* 24, 598–610. 10.1002/hipo.22252.
 10. Tsai, C.-H., Lin, B.-J., and Chao, P.-H.G. (2013). $\alpha 2\beta 1$ integrin and RhoA mediates electric field-induced ligament fibroblast migration directionality. *J. Orthop. Res.* 31, 322–327. 10.1002/jor.22215.
 11. Huang, L., Cormie, P., Messerli, M.A., and Robinson, K.R. (2009). The involvement of Ca²⁺ and integrins in directional responses of zebrafish keratocytes to electric fields. *J. Cell. Physiol.* 219, 162–172. 10.1002/jcp.21660.
 12. Zhu, K., Takada, Y., Nakajima, K., Sun, Y., Jiang, J., Zhang, Y., Zeng, Q., Takada, Y., and Zhao, M. (2019). Expression of integrins to control migration direction of electrotaxis. *FASEB J.* 33, 9131–9141. 10.1096/fj.201802657R.
 13. Yao, L., and Li, Y. (2016). The Role of Direct Current Electric Field-Guided Stem Cell Migration in Neural Regeneration. *Stem Cell Rev. Rep.* 12, 365–375. 10.1007/s12015-016-9654-8.
 14. Liu, J., Zhu, B., Zhang, G., Wang, J., Tian, W., Ju, G., Wei, X., and Song, B. (2015). Electric signals regulate directional migration of ventral midbrain derived dopaminergic neural progenitor cells via Wnt/GSK3 β signaling. *Exp. Neurol.* 263, 113–121. 10.1016/j.expneurol.2014.09.014.
 15. Ma, Y., Yang, C., Liang, Q., He, Z., Weng, W., Lei, J., Skudder-Hill, L., Jiang, J., and Feng, J. (2022). Direct Current Electric Field Coordinates the Migration of BV2 Microglia via ERK/GSK3 β /Cofilin Signaling Pathway. *Mol. Neurobiol.* 59, 3665–3677. 10.1007/s12035-022-02815-5.
 16. Myers, J.P., Santiago-Medina, M., and Gomez, T.M. (2011). Regulation of axonal outgrowth and pathfinding by integrin-ecm interactions. *Dev. Neurobiol.* 71, 901–923. 10.1002/dneu.20931.
 17. Hines, J.H., Abu-Rub, M., and Henley, J.R. (2010). Asymmetric endocytosis and remodeling of beta1-integrin adhesions during growth cone chemorepulsion by MAG. *Nat. Neurosci.* 13, 829–837. 10.1038/nn.2554.
 18. Robles, E., and Gomez, T.M. (2006). Focal adhesion kinase signaling at sites of integrin-mediated adhesion controls axon pathfinding. *Nat. Neurosci.* 9, 1274–1283. 10.1038/nn1762.
 19. Nichol, R.H., Hagen, K.M., Lumbard, D.C., Dent, E.W., and Gomez, T.M. (2016). Guidance of Axons by Local Coupling of Retrograde Flow to Point Contact Adhesions. *J. Neurosci.* 36, 2267–2282. 10.1523/JNEUROSCI.2645-15.2016.
 20. Eickholt, B.J., Walsh, F.S., and Doherty, P. (2002). An inactive pool of GSK-3 at the leading edge of growth cones is implicated in Semaphorin 3A signaling. *J. Cell Biol.* 157, 211–217. 10.1083/jcb.200201098.
 21. Mack, T.G.A., Koester, M.P., and Pollerberg, G.E. (2000). The Microtubule-Associated Protein MAP1B Is Involved in Local Stabilization of Turning Growth Cones. *Mol. Cell. Neurosci.* 15, 51–65. 10.1006/mcne.1999.0802.

Reviewer #3 (Remarks to the Author):

The present paper uses retinae isolated from E6 chick embryos to study the underlying mechanism of retinal ganglion cell (RGC) axonal guidance induced by an external electrical field. Previous studies have largely focused on chemical cues, thus gaining more insights into the mechanisms of electric axonal guidance is innovative and important. This study uses a series of manipulations involving integrin antibodies, Mn²⁺ application, inhibitors of GSK3 β signaling, calcium and microtubule imaging to establish a pathway that involves a calcium ion accumulation on the anodal side of the axon, followed by integrin inhibition and reduced microtubule stabilization.

The methods are generally described well, with a few exceptions. Whereas the idea for the proposed low calcium>integrin>microtubule stabilization mechanism is interesting, the provided evidence is not particularly strong. There are problems with the experimental design, quality and interpretation of the data. Furthermore, the writing is short of being publication quality. Several times, there is a lack of logic flow in the writing. A good example for this is the abstract.

Main concerns:

1. No introduction. Paper directly starts with results. The readers need to be introduced to the background of the study
2. The manuscript's abstract needs an overhaul change, statements are either unclear to the reader and/or they are superficial and exhibit a lack of connection from one line to the subsequent line.
3. A lot of the claims are not well supported by the images and the quantifications. See detailed comments below.

Specific comments:

1. Premise statement in the second sentence of the abstract "However, the mechanism of axon guidance during the early embryonic development remains elusive since the existence of long-range chemoattractants was denied¹⁻³" is incorrect. Only one of the three papers focused on RGCs and a specific cue, netrin-1, which is involved in short-range guidance of RGC axons exiting the eye. These papers did not generally deny the existence of long-range chemoattractants.
2. Abstract, line 4: "an electric field (EF) exists"
3. Abstract: "The inhibition of integrin-microtubule stabilization signaling abolished the EF effect. Since the anodal surface of an axon encounters EF-moved Ca²⁺ more frequently than the cathodal surface, the less inhibited integrin on the cathodal side would stabilize the more microtubules to trigger axon steering toward the cathode."
Both of these statements are not clearly supported by the results in this paper.
4. Abstract, last sentence "The present study points to electric axon guidance as the primary mechanism for axon orientation during central nervous system development." Is too strong.
5. It is unclear how fluorescence intensity can be used to measure axonal length. Did the author measure the average intensity in the images? The author should explain this quantification better in the methods section.
6. The number of retinas very low in number in certain experiments. For instance, on page 2 at the bottom, the experimenter mentioned only one retina used for the treatment of 2.0 mM Mn²⁺.
7. For several of the quantifications (Figure 1, 2, 4 and Extended Data Figure 1,7), the author should indicate not only the number of retinæ but also number of independent experiments. Does 1 retina in one condition equal 1 independent experiment? Are the data from 1 retina first averaged and then the averages of different retinæ are averaged?
8. Whereas the integrin antibodies TASC and W1B10 enhance growth with and without EF, from this data the author cannot conclude "Integrin mediates EF effects" as stated in the section title.
9. Whereas there are drawings of the microfluidic chip, the whole set up is difficult to visualize in the actual experiment. It would be good to include neurons and axons in the drawings.
10. The author combined a microfluidic chip with EF field (Ext Data Fig.4) and looked at axonal growth. However, a proper quantification of these axonal growth measurements with and without channel and with and without EF at different strengths is needed to make the conclusion that RGC axons are electrically directed to the future optic disc.
11. The quality of the Ca²⁺ imaging on either side of the axon in Fig. 3 is not very good. Where was the line scan measurement shown in panel d placed in panel c? Whereas there is a comparison of the anodal to cathodal side in panel g, no statistics has been done. Also, it is not clear why in this experiment the EF was positioned perpendicular to the axon.
12. In the microtubule stabilization experiments of Fig. 4, an essential control without inhibitor is

missing in order to make any statements that EF field effects are mediated by microtubule stabilization. As presented, the results in Fig. 4 do not provide any evidence that EF field-mediated growth involves microtubule stabilization.

13. The effect of SC79 is not clear, because no control is shown in panel Fig.4i.

14. The signaling pathway in the extended Fig. 8 is an interesting hypothesis; however, the evidence for it is not very strong. Instead of manipulating the pathway in the same direction as the EF is proposed to do, it would be better to interfere with the pathway, e.g. by inhibiting PI3 kinase or Akt instead of activating Akt or inhibiting GSK3beta. The author indeed inhibited PI3 kinase as shown in extended Fig. 7 but no effect was observed, which does not support the proposed signaling pathway.

15. The actin and microtubule labeling in panels j-o are not conclusive. A cathode-oriented higher concentration of microtubules is not obvious from the images shown. No quantification is shown. Throughout the paper, it would also help labeling the individual images better with the treatment or labeling probes.

16. Highly overstated final sentence: "The present study revealed the role of EF in orienting axons and pointed to electric axon guidance as the primary mechanism for axon orientation during CNS development." In terms of quantity and quality, the current literature has much more evidence for chemical than electrical axon guidance.

17. The statistics method section is titled with "Statics".

Re: COMMSBIO-22-3038-T

Responses to the comments of three reviewers:

The followings are my responses to the comments raised by the three referees. The parts that have been changed are written in blue in the revised manuscript.

Reviewer #1:

The main concerns of Reviewer #1 were the lack of a substantial introduction to explain the rationales for this study and the unclearness of the way to calculate the ventral extension rate. To address these concerns, the following changes were made in the revised manuscript. The parts that have been changed are written in blue in the revised manuscript.

Responses to Major Comments:

1. The lack of a substantial introduction/research background was pointed out. Introduction was written to explain the background and the rationales for this study in the revised manuscript.
2. The way to calculate the ventral extension rate (VER) was not fully explained. Reviewer #1 was concerned about the straightness of the ventral edge of retinal strips. To obtain VERs, the ventral edge of retinal strips should be straight. For this reason, I excluded the retinal strips of which the ventral edge was not straight. This is mentioned in Methods. It was also pointed out that axon outgrowth along the ventral strip is not consistent. Since the density of RGCs is high at the central region, numerous RGC axons appeared at the central part of retinal strips. This is mentioned in the first paragraph of Results. The size of retinal strips is described in Methods. The entire 150-400 μm rectangle parallel to the ventral edge was used to calculate VERs. The rectangle is indicated in the revised Fig. 1b and this is mentioned in the figure legend. Fluorescence images were taken at the level of RGC layer, which provided representative images. This is mentioned in Methods. Reviewer #1 suggested an additional set of in vitro experiments for tracking only a few axons in EFs. Instead of this type of experiments, I made a new set of experiments, the microchannel assay to quantify electric axon orientation. The results are presented in new Fig. 3 and Extended Data Fig.6.
3. The reason for looking into integrin is fully explained in Introduction. Reviewer #1 kindly suggested that it might be helpful to show how the loss of integrin impacts RGC axon

growth via knockdown experiments. Randlett and collaborators have done this experiment (Ref. 12) and their result is described in Introduction.

4. The rationale for detecting asymmetry of Ca^{2+} is explained in Introduction and at the beginning of “Asymmetry of EF-moved Ca^{2+} ” in Results.

Responses to Minor Comments:

1. The sentence “the existence of long-range chemoattractants was denied” in Abstract was deleted.
2. “that” was replaced with “and”.
3. A section title was added to the first section of Results.
4. “the” before “more” was deleted.
5. “the” was put before “retina”.
6. The sentence “the existence of long-range chemoattractants was denied” in Discussion was deleted

Reviewer #2:

Reviewer #2 raised three concerns to better support the conclusions: 1) the contribution of integrin to the orientation of axons in response to EFs needs to be strengthened, 2) the role of integrin as well as 3) other aspects of the mechanistic model need to be clarified.

The parts that have been changed are written in blue in the revised manuscript.

Responses to the detailed comments about the three concerns:

1. To directly test the ability of EF to control axon orientation and to quantify axon convergence in focused EFs, a new setup of experiments was made. A microchannel chamber was designed to orient RGC axons toward the microchannel. The convergence angle (CA), the total fluorescence intensities within CA (F_{CA}), and the mean fluorescence intensities within CA (mean F_{CA}) were introduced as parameters of axon convergence in new Fig. 3. Reviewer #2 kindly suggested that comparing trajectory defects due to the loss of EF by blocking epithelial Na^+ channels of retinal neuroepithelial cells (Ref. 7) to the ones due to Mn^{2+} addition (Fig. 2k) would help to support the contribution of $\beta 1$ integrin to EF-mediated guidance in whole retina. Following this important suggestion, the previous and present results were compared in Results.
2. I used two antibodies against $\beta 1$ integrin (TASC and WB10). According to the literature, TASC promotes adhesion to laminin, whereas WB10 inhibits it. Despite of such opposite

effects, the two antibodies enhanced EF-induced cathodal extension of RGC axons and also the basal extension without EF. To clarify the net enhancement by the antibodies of EF-mediated ventral growth, EF index (EFI) was introduced (new Extended Data Fig. 1e). The EFIs indicate the net enhancement of EF effects by the two antibodies as compared to the control without the antibodies. Reviewer #2 requested clear explanation of the effects of the two antibodies. To address this concern, two roles of microtubules in axon extension are assigned to the two antibodies in the revised Discussion; an axon is extended by two ways, 1) axon shaft elongation by stable microtubules and 2) growth cone advance by dynamic microtubules. It was thought that TASC enhanced the former process, while W1B10 enhanced the latter. Thus, the two antibodies are likely to contribute to axon extension in different ways. Staining patterns of the two antibodies would be interesting, but this experiment will be a new project. As another interfering tool, the RGD peptide was suggested, but it does not act as a ligand for integrin $\alpha6\beta1$ (laminin receptor). Another concern about integrin function was the concentration of Mn^{2+} (500 μM) used to maximize the affinity of integrin for the ligand. Since the high concentration of Mn^{2+} is toxic for hippocampal neurons, to test a lower concentration (100 μM) was suggested. Following this suggestion, Mn^{2+} at 100 μM was applied to the microchannel assay and it decreased EF effects (new Fig. 3c. d-f). Mn^{2+} at 500 μM completely abolished EF effects (Extended Data Fig. 6h). Fig. 2j shows that the RGC axon trajectory in the central region with 500 μM Mn^{2+} was the same as the control without Mn^{2+} . This fact could exclude the toxic effect of 500 μM Mn^{2+} . This is mentioned in Results.

3. To monitor extracellular Ca^{2+} , I used CalbryteTM-520L (potassium salt), not CalbryteTM-520L-AM, which is cell-permeable and used to monitor intracellular Ca^{2+} . This was a simple misunderstanding of Reviewer #2. It was suggested that to put the Calbryte-containing micropipette on the cathodal side could be an important control. This control experiment had been done and the result was added to Results and is presented in new Extended Data Fig. 8. To measure Ca^{2+} fluorescence on a growth cone was suggested. However, Ca^{2+} fluorescence was measured on an axon shaft because fine protrusions of a growth cone could not be identified with transmitted light. The reason for this choice was explained in the revised Results. The difference between anodal and cathodal sides in integral values of Ca^{2+} fluorescence intensities was statistically described in the revised Results. The statement on this quantitative difference was added to Discussion. Immunolabeling of $\beta1$ integrin and GSK3 in growth cones could be interesting, but this experiment will be a new project. Quantification of asymmetric distribution of stable microtubules was suggested to support the model for axon steering. Following this suggestion, the asymmetry index (AI) was introduced in new Fig. 5 and Extended Data Fig. 10.

The rationales for testing integrin are fully explained in Introduction by citing the suggested references on the role of integrin in EF-oriented cell migration. How integrin could regulate microtubules is better explained in the revised Discussion. The other signaling pathways (beside CRMP2 and APC) that GSK3 could regulate were introduced by citing the suggested references on EF-induced cell migration (Refs. 50,51). The roles of integrin in growth cone responses were also discussed by citing the suggested references (Refs. 38-43). The regulation of GSK3 activity in growth cones and the contribution of MAP1B to the turning of RGC growth cones were stated as the background for testing an inhibitor of GSK3 β (Refs. 33,34). The role of asymmetric PIP3-Akt signaling is discussed by adding a reference (Ref. 47). To make the statement less definitive, “The present study revealed the role of EF in orienting axons and pointed to electric axon guidance as the primary mechanism for axon orientation during CNS development” was deleted. Extended Data Fig. 4h from the paper by Dominici *et al.* (Ref. 2) was correctly explained in the revised Discussion. Finally, *P* values are described in Results or figure legends. The previous Fig. 3g was remade not to compare the mean values of the paired two samples (anodal and cathodal). Instead, the ratio (anodal/cathodal) was plotted against the cathodal value for each axon ($n = 12$) in new Fig. 4g.

Reviewer #3:

Reviewer #3 commented that gaining more insights into the mechanisms of electric axonal guidance is innovative and important since previous studies have largely focused on chemical cues. Reviewer #3 raised three main concerns and seventeen specific comments. The parts that have been changed are written in blue in the revised manuscript.

Responses to the main concerns:

1. No introduction. \rightarrow Introduction was written to explain the background.
2. Abstract needs an overhaul change. \rightarrow Abstract was overhauled.
3. The quantifications of images. \rightarrow The asymmetry index was introduced in new Fig. 5.

Responses to the specific comments:

1. The sentence in the previous abstract, “the existence of long-range chemoattractants was denied¹⁻³” was deleted.
2. “the electric field” was changed to “an electric field”.
3. The statement in the previous abstract, “The inhibition of integrin-microtubule stabilization signaling abolished the EF effect. Since the anodal surface of an axon encounters EF-moved Ca²⁺ more frequently than the cathodal surface, the less inhibited integrin on the cathodal

side would stabilize the more microtubules to trigger axon steering toward the cathode” was deleted in the revised Abstract.

4. The last sentence in the previous Abstract, “The present study points to electric axon guidance as the primary mechanism for axon orientation during central nervous system development” was deleted because it was too strong.
5. The way to calculate the ventral extension rate (VER) was fully explained in the revised Methods and the rectangle of the area for fluorescence measurement is indicated in the revised Fig. 1b.
6. Only one retina was used for the treatment of 2.0 mM Mn^{2+} . This result was deleted because the number of retina was small.
7. For data quantifications, not only the number of retinæ but also the number of independent experiments should be indicated. A question was raised: “Does 1 retina in one condition equal 1 independent experiment?” Yes, the number of retinæ equals the number of independent experiments. This and the way of averaging data from multiple retinæ are described in “Data collection and statistics” in the revised Methods.
8. Reviewer #3 commented that since the integrin antibodies enhanced axon growth with and without EF, it could not be concluded that “Integrin mediates EF effects”. To address this issue, the EF index (EFI) was introduced. The EFI indicates the difference in VER between no EF and with EF. Despite the fact that the antibodies enhanced the basal growth without EF, EFIs were significantly larger with the antibodies than the control without them. These results are presented in new Extended Data Fig. 1e.
9. The whole setup of the fan-shaped microfluidic chip is shown in new Extended Data Fig. 3a. RGC axons are included in the drawing as suggested by Reviewer #3.
10. A proper quantification of RGC axons in focused EF was needed to make the conclusion that RGC axons are electrically directed. To address this issue, the microchannel assay was made. The convergence angle (CA), the total fluorescence intensities within CA (F_{CA}), and the mean F_{CA} were introduced as parameters of axon convergence (Extended Data Fig. 6). These results are presented in new Fig. 3.
11. The presentation of the results of Ca^{2+} imaging was improved according to the comments of Reviewer #3 (new Fig. 4). The line scan data in panel c was added to panel d. The revised panel d also shows the transverse profile of the whole area. This is mentioned in the figure legend. The comparison of the anodal to cathodal sides was statistically described by using the ratio of anodal/cathodal in the revised Results and new Fig. 4g. The reason why the EF was positioned perpendicularly to the axon is also mentioned in Results.
12. In the blocking experiments of the signaling pathway for microtubule stabilization, an essential control without inhibitor was needed. To address this comment, the EF index (EFI) was introduced. The EFI of the control without inhibitors and EFIs with the drugs are presented in new Fig. 5j.

13. The effect of SC79 is clearly shown in new Fig. 5j with the control EFI.
14. It was pointed out that the inhibition of PI3K is necessary to support the proposed signaling pathway. The result of PI3K inhibition is presented in Extended Data Fig. 11 with the control EFI. This result is discussed in the revised Discussion. To test not only the activator (SC79) but also an inhibitor of Akt was suggested to support PI3K-Akt pathway. Since Akt is a suppressor of apoptosis (Biochem. J. 415: 333-344, 2008, Ref. 37), the inhibition of Akt by applying MK-2206 caused an apoptotic morphological change and the loss of axons. These results are shortly mentioned in Results.
15. To quantify asymmetric distribution of microtubules, the asymmetry index (AI) was introduced (new Fig. 5m and Extended Data Fig. 10). The mean AI was statistically described in Results. The labeling probes are indicated for individual fluorescence images following the kind suggestion (new Fig. 5l, o).
16. The last sentence in the previous Discussion, “The present study revealed the role of EF in orienting axons and pointed to galvanotropism as the primary mechanism for axon guidance during CNS development.” was deleted because it was overstated.
17. The statistics (not “Statics”) process was rewritten in the revised Methods (Data collection and statistics).

Reviewers' comments:

Reviewer #1 (Remarks to the Author):

The authors have addressed all of my major concerns.
Overall, the current manuscript has gained clarity.

Reviewer #2 (Remarks to the Author):

Dear author,

We thank the author for taking in consideration most of the comments. To me, the manuscript has been greatly improved. First, the context and questions are better introduced and some points have been clarified in the discussion. Second, several additional quantifications, control and analyses have been made that strengthen the results. Finally, as state in my previous review, I think that the present work could be of interest for the community and rises interesting hypothesis on the way EFs could provide spatial information to axons.

We thank the author for:

1-a, providing a new set of experiments that includes quantification of axon orientation and convergence in response to EF.

1-b, mentioning his previous results on axon orientation in the dorsal retina after inhibition of endogenous EF.

2-a, stating the different impacts of the TASC and W1B10 antibodies on integrin activity and providing an explanation for their similar effect on outgrowth and EF-response.

2-b, calculating the EF specific responses (EFI), which help to clarify the effects of the antibodies and Mn⁺⁺ in presence of EF.

2-c, including an experiment done at lower concentration of Mn⁺⁺ and highlighting observations that indicate that Mn⁺⁺ effects are not due to general deleterious effects on neurons.

3-a clarifying the Calbryte-520L used.

3-b, adding an experiment with the reverse orientation for Ca⁺⁺ measurement.

Nevertheless, it is not clear to me whether similar anode/cathode ratio is found in the "reverse" condition.

3-c, presenting the anode/cathode ratio of Calbryte signal changes upon EF stimulation. This gives clearer view of the differences. This is important as it could normalize differences in basal fluorescence.

However, it is unclear to me whether it could correct the differences between anodal and cathodal fluorescence without EF (Fig4e and f).

3-d, including a quantification of the asymmetric distribution of stable microtubules.

We agree that:

1-b, choices have to be made, but I regret that the author did not provide quantification of the intraretinal guidance defects as:

-the image (Fig. 2k) is not easy to interpret (difficult to see the changes of orientation and that it seems that axons density decreases)

-this experiment relates more directly to in vivo situation.

-it would be easier to compare phenotypes with quantitative parameters.

2-a, immunolabeling for TASC and W1B10 might part of new project.

2-b, finding an additional way to block $\alpha 6\beta 1$ integrins cannot be done for the revision as RGD peptide did not bind to these integrins and that siRNA knock-down would have been time consuming.

3-d, immunolabeling of integrin and GSK3 are not essential to support the conclusions for this manuscript.

I have comments that the author might want to consider to further improve the manuscript:
1-a, use a different image for the 100 μ M Mn⁺⁺ that could illustrate better the inhibition of EF-mediated response (Fig.3c). Better explain how axons were selected to calculate the angle (Fig.3, extended fig 6 and "material and method")

1-b, change the image (fig2.k) to better illustrate the misorientation of axons or point toward abnormally oriented axons.

3-a, put the reference number and the distributor of Calbryte-520L in the material and method as done for other reagents. check that this is the case for all key reagents.

3-b, state the number of experiments done in the reverse condition.

3-c, show the anode and cathode ROI on the axon overview (fig4 a or b). Use the same scale for Fig4 e and f. calculate anodal and cathodal fluorescence ratio before EF application.

Although the author changed the specific sentence I pointed out in the discussion, I feel that other statements could be more nuanced. For example, it seems to me that claiming that "the guidance cues for commissural axons are unknown" is too strong. Indeed, even if the role of Netrin has been questioned, it still contributes to guide the axons ventrally and other cues such as Shh can attract the axons to the floor plate. Furthermore, different cues have been shown to control pre-crossing trajectory as well as floor plate crossing and post-crossing navigation (1–4). Yet, even in this model that has been studied a lot, our knowledge is still too limited to fully explain axon navigation and there is room for other cues.

I also understand that identification of growth cone could be difficult in the present set-up. However, axon trajectories are considered to be primarily shaped by growth cones behaviors. Thus, even if it is important to show that some observations made on the axon shaft could apply to growth cone. It is also important to keep in mind that this not necessarily always the case. Indeed, as stated by the author, the two structures have different microtubule dynamics (lines 281-283). Thus, the authors could be more careful in some sentences such as the one lines 264-265.

Statistical Analysis:

Finally, I thank the author for providing better description of the statistical analyses done (indication of the p value, description of the tests used and number of samples).

The authors should justify the choice made. T-tests might not be the most appropriated as it is difficult to apply when sample numbers are too low to check normality. The author should explain why one-tailed tests that are based on a prediction of the effect, were used instead of two-tailed tests for some conditions.

As t-tests were done, the authors could add lines above histograms bars to explicitly show which comparisons were made and facilitate the reader understanding.

1. Alvarez, S., Varadarajan, S.G., and Butler, S.J. (2021). Dorsal commissural axon guidance in the developing spinal cord. In *Current Topics in Developmental Biology* (Elsevier), pp. 197–231. 10.1016/bs.ctdb.2020.10.009.
2. Ducuing, H., Gardette, T., Pignata, A., Tauszig-Delamasure, S., and Castellani, V. (2019). Commissural axon navigation in the spinal cord: A repertoire of repulsive forces is in command. *Semin. Cell Dev. Biol.* 85, 3–12. 10.1016/j.semcdb.2017.12.010.
3. Moreno-Bravo, J.A., Roig Puiggros, S., Mehlen, P., and Chédotal, A. (2019). Synergistic Activity of Floor-Plate- and Ventricular-Zone-Derived Netrin-1 in Spinal Cord Commissural Axon Guidance. *Neuron* 101, 625-634.e3. 10.1016/j.neuron.2018.12.024.
4. Chédotal, A. (2019). Roles of axon guidance molecules in neuronal wiring in the developing

spinal cord. Nat. Rev. Neurosci. 20, 380–396. 10.1038/s41583-019-0168-7.

Reviewers #3-4 (Remarks to the Author):

The author has addressed most of my concerns. He added new data and quantification and improved the manuscript. I have a few comments for further improvement.

Major concern:

I am still not convinced that he can claim an asymmetric MT distribution in response to the electric field. There is now more data shown on MT quantification in extended Figure 10; however, it is unclear whether the approach of choosing the red center line of RGC axons is appropriate to determine the center of the axon. Based on the quality of the images in both transillumination and fluorescence as shown, the asymmetric MT distribution is not convincing to this reviewer. To claim this asymmetric stabilization better data is needed. One possibility is comparing the tubulin signal with a membrane dye. The membrane marker should clearly delimit the border of the axon. Another possibility would be total tubulin vs. acetylated tubulin immunostaining. With the current data, I do not believe that the author can claim asymmetric MT stabilization. It could be hypothesized but not claimed.

Minor concerns:

1. I suggest changing the title to "Integrin-mediated electric axon guidance for optic nerve formation"
2. Some of the technical descriptions about new results should go into the methods section, e.g. for example some of the technical details in the new results section "Integrin mediates RGC axon convergence".
3. The author included data with inhibition of phosphatidylinositol-3 kinase (PI3K) in extended data Fig. 11 in the discussion; however, it is unclear why that was not shown in the last result section.
4. It would help to have the direction of the electric field indicated on the images of extended data Fig. 10.
5. Line 802: it should say "a range of 0-255"
6. Extended Data Fig. 6: "mean" not "meam" in panels d-f.

Re: COMMSBIO-22-3038B

Responses to the comments of Reviewer #2 and Reviewers #3-4:

The followings are my responses to the comments of Reviewer #2 and Reviewers #3-4. All changes are highlighted in blue in the revised manuscript text file.

Reviewer #2:

I appreciate helpful comments on the manuscript. To address the comments, the following changes were made in the text and figures. All changes are highlighted in blue in the revised manuscript text file.

Point-by-point responses:

1-a, to better illustrate the inhibition by Mn^{2+} of EF-mediated axon convergence, a different image was used in Fig. 3c and Extended Data Fig. 6c. The explanation for the selection of axons for convergence angle measurement was added to Methods (Microchannel assay).

1-b, yellow arrowheads were added to Fig. 2k to indicate abnormal axon trajectories.

3-a, product number of Calbryte-520L and details of other reagents were added in Method.

3-b, the number of axons tested with the reverse EF for Ca^{2+} measurement was added to Results. The cathodal/anodal ratios of integral values of decreases in fluorescence intensity during the reverse EF were smaller than the mean cathodal/anodal ratio of the fluorescence intensities before EF application (Extended Data Fig. 8c). This could imply that Ca^{2+} was blown off by the reverse EF without supply of Ca^{2+} from the anodal side. On the other hand, the integral values of fluorescence increase during the forward EF exceeded the mean fluorescence ratio before EF application (Fig. 4g).

3-c, the anode and cathode ROIs were enlarged (insets in Fig. 4e and f). The ROIs could not be superimposed on transmission images (Fig. 4a or b) since a ROI was set on a fluorescence image. The same scale was used in Fig. 4e and f to show the difference in the fluorescence intensities before EF application. The mean anodal/cathodal ratio of the fluorescence intensities before EF application was added to Results and Fig. 4g.

Reviewer #2 pointed out that “the guidance cue for commissural axons is unknown” was too strong. This sentence was changed to “the guidance cues for commissural axons are still enigmatic”. The four references suggested by Reviewer #2 were added to Discussion (60-63).

The sentence in lines 264-265 was carefully rewritten and changed to “In a turning axon, microtubules are stabilized on the side to turn in the growth cone and growing shaft^{30,48}. The asymmetric microtubule stabilization would steer the axon toward the cathode”.

Statistical Analysis:

Following the helpful comment of Reviewer #2, two-tailed *t*-test was applied to all of the comparisons. Adding lines above histograms bars was also suggested to indicate which comparisons were made. Instead of the horizontal lines between bars, histograms bars were displayed in different colors when they had a significant difference. The bars in the same color indicate no difference between them. Usages of colors are explained in figure legends.

Reviewers #3-4

Major concern:

The main concern of Reviewer #3 was an asymmetric MT distribution. The MT fluorescence shows that the peak MT fluorescence is cathodally shifted at the half level of MT fluorescence. The red vertical line in Extended Data Fig. 11 (previous 10) indicates the center of the half level of MT fluorescence, not the exact axon center as Reviewer #3 was concerned about. However, since the length of the half level of MT fluorescence is approximately 100 pixels and one pixel size is 14.4 nm, the half level of MT fluorescence almost covers the diameter of an axon. An asymmetric MT distribution was found at a turning point, not found at straight parts (Extended Data Fig. 11b). Thus, MTs are likely to be asymmetrically distributed at a turning point of an axon in response to an EF. Reviewer #3 suggested double fluorescence imaging (membrane marker and MT fluorescence) and immunostaining. However, the double fluorescence imaging requires two sets of filters and a quick exchanger of filters to record from live axons. The immunostaining study would provide important data concerning MT distribution. However, this study requires a high magnification confocal fluorescence microscope equipped with a multiple wavelength

system or an electron microscope, and will be a new project.

Minor concerns:

1. Following the kind suggestion, the title was changed to “Integrin-mediated electric axon guidance for optic nerve formation”.
2. The technical details regarding axon convergence in focused EFs (Fig. 3) were moved from Results to Methods.
3. The results of PI3K inhibition were added to Results.
4. The direction of EF was indicated in the images of Extended Data Fig. 11 (previous 10).
5. Line 802: “0-256” was corrected to “0-255”.
6. Extended Data Fig. 6: “meam” was corrected to “mean” in panels d-f.